# Large-language models facilitate discovery of the molecular signatures regulating sleep and activity

Di Peng [1,4], Liubin Zheng [1,4], Dan Liu [1,4], Cheng Han [1,4], Xin Wang[1], Yan Yang[1], Li Song[1], Miaoying Zhao[1], Yanfeng Wei[1], Jiayi Li [1], Xiaoxue Ye[1], Yuxiang Wei[1], Zihao Feng [1], Xinhe Huang[1], Miaomiao Chen [1], Yujie Gou[1], Yu Xue [1,2] ✉ & Luoying Zhang [1,3] ✉

Sleep, locomotor and social activities are essential animal behaviors, but their reciprocal relationships and underlying mechanisms remain poorly understood. Here, we elicit information from a cutting-edge large-language model (LLM), generative pre-trained transformer (GPT) 3.5, which interprets 10.2–13.8% of *Drosophila* genes known to regulate the 3 behaviors. We develop an instrument for simultaneous video tracking of multiple moving objects, and conduct a genome-wide screen. We have identified 758 fly genes that regulate sleep and activities, including *mre11* which regulates sleep only in the presence of conspecifics, and *NELF-B* which regulates sleep regardless of whether conspecifics are present. Based on LLM-reasoning, an educated signal web is modeled for understanding of potential relationships between its components, presenting comprehensive molecular signatures that control sleep, locomotor and social activities. This LLM-aided strategy may also be helpful for addressing other complex scientific questions.

Animals rarely act in isolation. Their interactions with conspecifics exert heavy influences on their behavior and physiological state. Although the idea that quantity and quality of social relationships are major risk factors for health both in humans and other animals was proposed over 30 years ago, the underlying mechanisms remain largely unclear[1]. Fruit flies are social animals that display dynamic social interaction networks and collective behaviors that contribute to a variety of processes essential for life[2,3]. Prior social experience modifies sleep needs and architecture, while chronic social isolation leads to sleep loss and increased feeding activity[4,5]. Sleep and locomotor activity of the fly population are also distinct from that of isolated individuals, but how sleep and locomotor activity are regulated in a group setting with the presence of conspecifics and how they interact with social activities remain to be characterized[6].

Compared to other artificial intelligence (AI) techniques, the main purpose of natural language processing (NLP) is to learn, interpret and generate human language content[7]. Recently, great breakthroughs have been achieved regarding generative artificial intelligence (GAI) based on large-language models (LLMs)[8,9], especially generative pre-trained transformer (GPT) and its updates[10,11]. Interestingly, it has been demonstrated that GPT possesses the features of general-purpose technologies, exhibiting the potential to revolutionize labor productivity[12]. In order to develop GPT-3, the architecture of Transformer neural network using an attention or self-attention mechanism was adopted[13]. For model training, GPT-3 used a combination of five datasets, including filtered monthly CommonCrawl data covering 2016–2019, an expanded release of the WebText dataset (WebText2), two Internet-based books corpora (Books1 and Books2) and English-

[1]Key Laboratory of Molecular Biophysics of Ministry of Education, Hubei Bioinformatics and Molecular Imaging Key Laboratory, College of Life Science and Technology, Huazhong University of Science and Technology, Wuhan, Hubei 430074, China. [2]Nanjing University Institute of Artificial Intelligence Biomedicine, Nanjing, Jiangsu 210031, China. [3]Hubei Province Key Laboratory of Oral and Maxillofacial Development and Regeneration, Wuhan, Hubei 430022, China. [4]These authors contributed equally: Di Peng, Liubin Zheng, Dan Liu, Cheng Han. ✉e-mail: xueyu@hust.edu.cn; zhangluoying@hust.edu.cn

language Wikipedia, approximately equivalent to 500 billion byte-pair-encoded tokens[14]. Later, GPT-3 was fine-tuned into the InstructGPT model by conducting reinforcement learning from human feedback (RLHF)[15,16]. Besides the InstructGPT dataset, a dialog dataset, not clearly reported by OpenAI, was generated from conversations between human AI trainers and the chatbot, which was further fine-tuned to the state-of-the-art model, ChatGPT or formally GPT-3.5 (https://openai.com/blog/chatgpt). To solve complex tasks, prompt engineering has been demonstrated as an effective strategy for eliciting information from LLMs[14,17]. In particular, complex reasoning in LLMs can be elicited by using a method termed chain-of-thought (CoT) prompting[17]. Although machines are capable of manipulating and producing language knowledge[10], how LLMs can be employed to facilitate scientific research is largely unclear[18].

Here, we first conduct a genome-wide interpretation of the genetic basis of sleep, locomotor, and social activity regulation in *Drosophila melanogaster*, using a standard prompting strategy to elicit information from GPT-3.5. 12.5%, 13.8%, and 10.2% of the fly protein isoforms are interpreted to be involved in sleep, locomotor, and social activity regulation, respectively, with low false positive rates. These numbers demonstrate the usefulness of GPT-3.5 in the collection and summarization of relevant information. In parallel, we develop a video-tracking instrument to simultaneously monitor the real-time behavior of multiple fruit flies. Using this system, we conduct a genome-wide RNA interference (RNAi) screen and identify 285, 310, and 359 genes to be potentially involved in regulating sleep, locomotor, and social activity, respectively. Besides a number of genes recognized by GPT-3.5 to regulate these three behaviors based on published literature, we also identify many more that have not been previously reported, such as *mre11* and *NELF-B*. Then, an educated signaling network is modeled for 86 candidates from the screen, using the CoT prompting strategy of LLM-reasoning to pairwisely identify potential functional regulations or associations among these genes. We further validate the potential mechanisms reasoned by LLM and reveal that MRE11 may influence sleep, locomotor, and social activity by modulating dopamine receptor *Dop1R1* and *Histidine decarboxylase* (*Hdc*). In summary, here we systematically analyze the molecular mechanism regulating sleep, locomotor, and social activities by utilizing in silico interpretation and reasoning from LLMs-generated contextual information, in combination with genetic screens using our multi-object video-tracking paradigm. We anticipate that such a human-LLM interactive practice can be readily adapted to investigate other complex scientific questions as well.

## Results

### Genome-wide LLM interpretation and development of a video-tracking instrument to investigate the genetic basis of sleep, locomotor, and social activities

First, we explored the genetic basis of sleep, locomotor, and social activity regulation by querying LLMs whether a protein isoform is functionally involved. To ensure the efficiency of prompt-generation response[19], we used a standard prompting strategy, which incorporates the background knowledge step by step into the full prompts to elicit GPT-3.5 to produce high-quality answers. In total, we queried GPT-3.5 using the full names of 42,794 fly protein isoforms across the entire genome, and 128,382 prompt-generation pairs were produced by GPT-3.5 for interpreting the potential role of each protein isoform in regulating any of the three behaviors (Fig. 1a; Supplementary Data 1). In our results, we found LLMs were able to efficiently distinguish knowns from unknowns regarding protein functions, and the positive response rates are 12.5%, 13.8%, and 10.2% for genome-wide interpretation of knowledge regarding sleep, locomotor, and social activity, respectively (Fig. 1b; Supplementary Data 1).

To critically evaluate the performance of GPT-3.5 for interpreting the 3 behaviors, we manually curated experimentally identified genes

essential for sleep, locomotor, and social activity from the scientific literature (Supplementary Data 2a–c). In comparison with the prediction results of GPT-3.5, a standard measurement, sensitivity (Sn), was calculated to estimate the false negative rate (Type II errors). To estimate the false positive rate (Type I errors), an equal number of genes were randomly selected from genes that have not been reported to regulate each of the behavior, and the average specificity (Sp) value was calculated after 20 rounds of resampling. For sleep, locomotor, and social activity, the Sn values were 20.9%, 25.1%, and 18.4%, respectively (Fig. 1c; Supplementary Fig. 1a–c), showing a high false negative rate of GPT-3.5 responses. This reflects the limitations of GPT-3.5 in searching for information of this sort. Notably, the knowledge used for GPT-3.5 model training was limited to data before 2022 which also contributes partially to the high false negative rate. Indeed, an important sleep-regulating gene reported in 2022, *discs overgrown* (*dco/dbt*)[20], was not recognized as a positive hit by GPT-3.5 (Supplementary Data 1a and 2a). For the 3 behaviors, the average Sp values ranged from 92.8–92.9% (Fig. 1c; Supplementary Fig. 1a–c), showing a low false positive rate in GPT-3.5 answers. Despite the high false negative rate, we believe this low false positive rate still supports the usefulness of LLMs in searching and summarizing literature.

For the positive responses, the machine CoTs of LLMs were visualized in a word cloud, and we found that experimental strategies including genetic manipulations such as gene knockdown and over-expression, behavior monitoring, measurement of neuronal activation, as well as gene expression analysis are overrepresented (Fig. 1d; Supplementary Data 3). Taken together, these findings demonstrate that before starting a research project, GPT-3.5 can serve as a helpful tool to efficiently collect and summarize the relevant information with a low false positive rate, and provide feasible suggestions regarding research designs.

In parallel, we developed an instrument for multi-object tracking in videos, which can record the real-time behavior of fruit flies as two-dimensional coordinates of individual flies in a culture plate for days (Fig. 1e). In this way, we can monitor sleep and locomotor activity in a social context and examine their relationships with social activities. This paradigm comprises of an incubator with camera recording systems, fly culture plates, as well as the temperature and light control panel (Fig. 1e). Due to the large size of raw video files, we designed a video-to-location (VTL) strategy to transform raw videos into much more compact text files (.vtl) that record the two-dimensional coordinates of individual flies in the culture plate. The recorded videos were converted into frame-by-frame image slices using a media transcoder, OpenCV (https://opencv.org). Movements of each fly were calculated by the alteration of its location in each image frame, which was transformed to a data point in a two-dimensional coordinate system (Fig. 1f).

To evaluate the accuracy of this system for object tracking of individual flies, we monitored male and female flies with different group sizes ranging from 1 to 20, and 153,360 frames derived from 144 randomly selected video fragments (7.2 frames/s) were manually checked to calculate the accuracy (Supplementary Fig. 2a–c; Supplementary Data 4). We found that the accuracy for group sizes from 1 to 5 is ≥98.28%, while the accuracy is significantly decreased when the group size exceeds 5 animals (Fig. 1g; Supplementary Fig. 2a, b). Therefore, in this study, we used a group size of 5 unless specified otherwise. In these recordings, we found 3 types of tracking errors, including deviation, overlap, and mislocation (Fig. 1h–j; Supplementary Movies 1–3).

We next analyzed sleep, locomotor, and social activity of flies in isolation and in group sizes ranging from 2–5. As can be seen, when there are more than 3 individuals in a group, sleep duration is significantly reduced (Supplementary Fig. 3a, b). Sleep also becomes more fragmented with shortened sleep bout duration and increased sleep bout number, accompanied by decreased latency to fall asleep

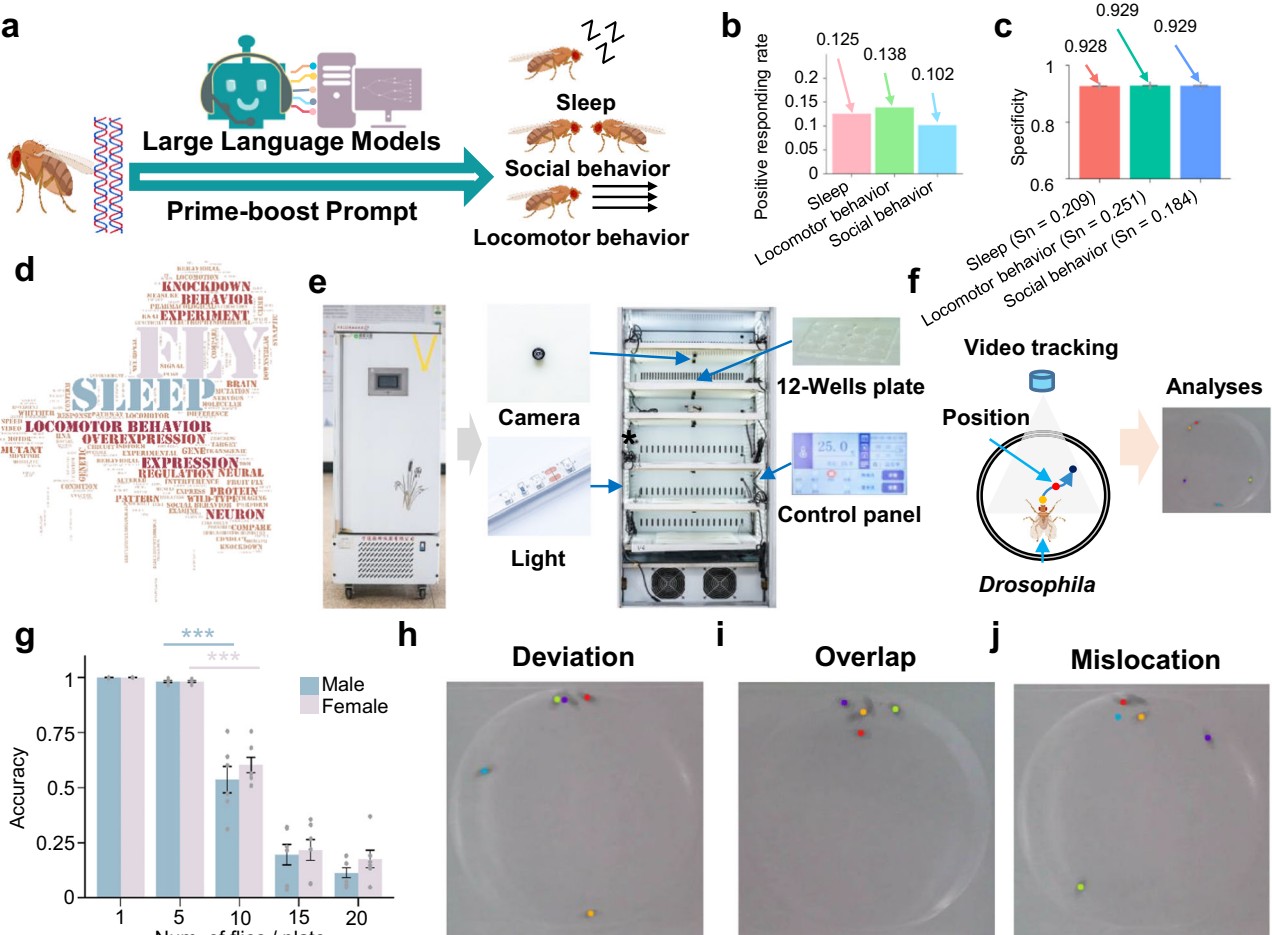

**Fig. 1 | Genome-wide LLM interpretation and development of a video-tracking instrument to investigate the genetic basis of sleep, locomotor, and social activity. a** A flowchart of genome-wide knowledge interpretation of 3 fly behaviors using the standard prompting strategy. The icons of flies, DNA, and computer are provided by Prof. Jian Ren. **b** Bar graph demonstrating the positive response rate of all prompt-generation pairs. **c** Bar graph demonstrating the prediction performance of prompt-generation pairs. (*n* = 20 independent experiments for the calculation of average Sp value). **d** A word cloud of experimental design used to study the 3 fly behaviors. **e** Photos demonstrating components of the video-tracking instrument. **f** Illustration of VTL strategy for measuring fly movements. The icon of fly is provided by Prof. Jian Ren. **g** Bar graph demonstrating the accuracy of movement tracking in group sizes ranging from 1 to 20 flies (*n* = 6 biologically independent experiments, one-sided Welch's *t*-test for unpaired comparisons, for male fly, fly number 5 *vs.* 10, \*\*\**p* = 0.00097, for female fly, fly number 5 *vs.* 10, \*\*\**p* = 0.00016). 3 error tracking patterns, including deviation (**h**), overlap (**i**), and mislocation (**j**). Data are presented as the mean ± SEM. Source data are provided as a Source Data file.

(Supplementary Fig. 3c–e). An increase of group size also substantially reduces locomotor and social activity (Supplementary Fig. 3f–h).

To test the validity of this system, we treated flies with various drugs that have been shown to alter sleep, social, and/or locomotor activity. Tyrosine hydroxylase inhibitor α-methyl-para-tyrosine (AMPT) which impairs dopamine synthesis increases sleep, while GABA-A receptor antagonist carbamazepine (CBZ) and agonist (THIP) reduces and enhances sleep, respectively, consistent with previous studies using isolated flies (Supplementary Fig. 4a, b)[21–23]. CBZ reduces sleep bout duration and sleep bout number, while AMPT and THIP lengthen sleep bout duration (Supplementary Fig. 4c, d). CBZ also enhances sleep latency while AMPT and THIP shorten sleep latency (Supplementary Fig. 4e). These drugs exerted effects on locomotor activity which is in the opposite direction to that of sleep (Supplementary Fig. 4f). In addition, AMPT treatment leads to a trend of reduction of social activity as previously reported (Supplementary Fig. 4g, h)[24].

Flies mutant for the gene encoding the receptor of neuropeptide pigment dispersing factor (*Pdfr^hanS304*) which are known to be long sleepers under isolated conditions also display increased sleep in the group condition, while flies mutant for gene encoding dopamine

transporter (*DAT^fmn*) are well-known short sleepers and exhibit decreased sleep in group condition (Supplementary Fig. 5a, b)[25–27]. In addition, *DAT^fmn* reduces sleep bout duration and bout number while *Pdfr^hanS304* prolongs sleep bout duration (Supplementary Fig. 5c, d). In line with the effects on sleep duration, *DAT^fmn* lengthens while *Pdfr^hanS304* shortens sleep latency (Supplementary Fig. 5e). *Pdfr^hanS304* is known to eliminate morning anticipatory activity and phase-advance evening anticipatory activity in isolated flies, and these alterations are also evident under group condition (Supplementary Fig. 5f, g)[28,29]. *Pdfr^hanS304* and *DAT^fmn* decreases and increases locomotor activity, respectively, similar to what has been observed in isolated flies (Supplementary Fig. 5h)[21]. Interestingly, significantly increased social activity is observed for *DAT^fmn* flies while decreased social activity for *Pdfr^hanS304* flies (Supplementary Fig. 5i, j). In addition, *Cullin 3* (*Cul3*), calcineurin A subunit *CanA-14F* as well as *Fmr1* which encodes fragile X messenger ribonucleoprotein 1 have all been reported to regulate sleep[30–32]. Here we also observed significant reduction of sleep duration when we knocked down *Cul3* and *CanA-14F* (Supplementary Fig. 6) or over-expressed *Fmr1* (Supplementary Fig. 7), consistent with previous studies. We further compared the sleep parameters of these flies in our group condition with the published work. We found a trend of

reduction in sleep bout length and lengthened sleep latency in *Cul3* RNAi flies, as well as shortened sleep bout length in *CanA-14F* RNAi flies, in agreement with what has been reported in the literature[30,32]. We observed decreased sleep bout duration and sleep bout number in *Fmr1* over-expressing flies, similar with the previous study[31].

For further validation, we activated octopaminergic neurons by expressing the temperature-gated depolarizing cation channel TrpA1 or the bacterial depolarization-activated sodium channel NachBac[33,34]. This manipulation has been shown to promote wakefulness and increase sleep latency in isolated flies, which are observed under group condition as well (Supplementary Fig. 8)[35].

Taken together, these results indicate that our system is capable of detecting alterations of sleep, social, and locomotor activity in group condition.

## Genome-wide RNAi screen to identify genes regulating sleep, locomotor, and social activities

Next, we employed our video monitoring system to identify genes that are involved in regulating sleep, locomotor, and social activity in the group condition by conducting a genome-wide RNAi screen using male flies. In total, we knocked down 5588 genes in all neurons using *elav*GAL4 to drive 6885 RNAi or mutant lines and monitored the movements of these flies under 12 h light/12 h dark condition for at least one day (Supplementary Data 5). The volume of raw video data was estimated to exceed 100 TB, and thus only VTL files were preserved after video recording and transformation.

We found that knocking down 285 genes leads to a short sleep with 2 standard deviations (SDs) lower than the population mean, including 20 known genes to be involved in sleep regulation based on our curated benchmark datasets (Fig. 2a; Supplementary Data 2a). Gene Ontology (GO)-based enrichment analysis demonstrates that biological processes, such as transcriptional regulation (GO:0045892, GO:0006357 and GO:0045944) and chromatin remodeling (GO:0006325 and GO:0031507), are enriched among these genes (Fig. 2b; Supplementary Data 6a). On the other hand, silencing 310 genes markedly increased locomotor activity by 50%, including 29 known genes were found to regulate locomotor activity (Fig. 2c; Supplementary Data 2b). Among these genes, transcriptional regulation (GO:0006357, GO:0000122, and GO:0045944), locomotor rhythm (GO:0045475), nucleolus organization (GO:0007000), and adult walking behavior (GO:0007628) are enriched (Fig. 2d; Supplementary Data 6b). Knocking down 359 genes reduced the social interaction duration by at least 50% compared to the population mean, among which 4 are previously known (Fig. 2e; Supplementary Data 2c). These genes are enriched in pathways associated with reactive oxygen (GO:0072593 and GO:0006979), mating behavior (GO:0007617), and the mitotic cell cycle (GO:0000278) (Fig. 2f; Supplementary Data 6c). Interestingly, Li et al. found that chronic social isolation induces a brain state that signals starvation, and we also observed an enrichment of genes involved in the cellular response to starvation (GO: 0009267)[5]. Among the positive hits, there are 52 genes that participate in regulating sleep and social activity, and 58 genes that regulate both locomotor and social activity, while 30 genes are involved in orchestrating all 3 behaviors (Fig. 2g). The identification of these genes may provide a starting point to unveil how sleep and locomotor activity interact with social activity.

## *mre11* promotes sleep only in the presence of conspecifics

We further characterized one of the top hits in our screen, *meiotic recombination 11* (*mre11*), a gene known to be involved in DNA damage repair and telomere protection and have not been reported to participate in sleep regulation[36,37]. Knocking down *mre11* with 2 independent RNAi lines both lead to dramatically reduced sleep duration in group but not under individual condition (Fig. 3a–d; Supplementary Fig. 9). Detailed analysis of sleep structure demonstrates that both

sleep bout duration and sleep bout numbers are much reduced, accompanied by lengthened sleep latency after lights off (Fig. 3e–g). These results indicate that lack of *mre11* impairs both sleep maintenance and initiation. Interestingly, *mre11* RNAi flies exhibit enhanced resistance to sleep deprivation, implying an elevated sleep pressure (Supplementary Fig. 10). We explored the minimum number of individuals required to induce the sleep decrease observed in group condition and found that 2 individuals are sufficient (Fig. 3h). Apparently, this reduction in sleep is not caused by increased social interaction as both the duration of social interaction and the number of social interaction bouts are decreased in *mre11* RNAi flies (Fig. 3i–l).

To investigate social factors that may contribute to this short-sleep phenotype, we first measured social space among these flies but observed no significant alteration (Supplementary Fig. 11a). However, *mre11* RNAi flies show substantial deficits in stress odorant response due to impaired avoidance of stress odor while the ability to emit stress odor appears intact, reflecting defective sensory processing of olfactory social cues (Fig. 3m; Supplementary Fig. 11b–d). Moreover, control flies demonstrate avoidance of an unpleasant odor (3-octanol, OCT) while this behavior response is much attenuated in *mre11* RNAi flies (Fig. 3n). On the other hand, *mre11* RNAi and control flies display comparable avoidance behavior when exposed to electric shock (Supplementary Fig. 12). Taken together, these results implicate that lack of *mre11* leads to defective olfactory sensation. It is possible that when conspecifics are present, this altered olfactory sensation enhances some social-related signals that strongly inhibit sleep in *mre11* RNAi flies. We suspected that perhaps *mre11* deficient flies incorrectly recognize conspecifics as heterospecifics, leading to sleep inhibition. To test this, we monitored each *mre11* RNAi fly and controls with another conspecific (*Drosophila melanogaster*) or a heterospecific (*Drosophila simulans*). However, daily sleep duration is substantially shortened regardless of the presence of a conspecific or a heterospecific, along with decreased sleep bout duration, suggesting that the sleep inhibition caused by *mre11* deficiency is not due to misrecognition of conspecifics as heterospecifics (Supplementary Fig. 13). Notably, the reduction of social interaction is no longer observed when a heterospecific is present (Supplementary Fig. 14). This is probably because social interaction is already much suppressed in control genotypes in the presence of a heterospecific. However, we noticed that the frequency of social interaction is increased in *mre11* RNAi flies in the presence of a heterospecific, implying that MRE11 may be involved in conspecific/heterospecific recognition.

In order to probe the mechanism by which *mre11* deficiency leads to shortened sleep under group condition, we first attempted to identify the anatomical substrate that mediates this effect by knocking down *mre11* in different brain structures and cell types that are involved in sleep regulation (Supplementary Fig. 15). *mre11* does not appear to be required at any of these anatomical sites for maintaining normal sleep, suggestive of unknown sleep regulatory mechanism involved. We then took a pharmacological approach and fed flies with drugs that impinge on different neurotransmitter systems (Fig. 3o; Supplementary Fig. 16). AMPT treatment eliminates the majority of the sleep differences between *mre11* RNAi and controls, which means the influences of MRE11 on sleep likely requires dopaminergic signaling (Fig. 3o). AMPT also abolishes (at least in part) the reduced social interaction duration and increased activity of *mre11* RNAi flies (Supplementary Fig. 17). In addition, we examined the effects of knocking down *mre11* on the expression of known sleep genes (Fig. 3p; Supplementary Fig. 18). The mRNA levels of several ion channels that promote sleep are significantly reduced compared to the GAL4 control, including *Hyperkinetic* (*Hk*), *Shaker cognate l* (*Shal*), *Shab* and *Ih* while no significant difference was observed for known wake-promoting genes[38–41] (Supplementary Fig. 18a, b; Supplementary Data 7). After including the UAS control, we found that *mre11* deficiency only decreases the expression of *Hk* and the sleep-promoting

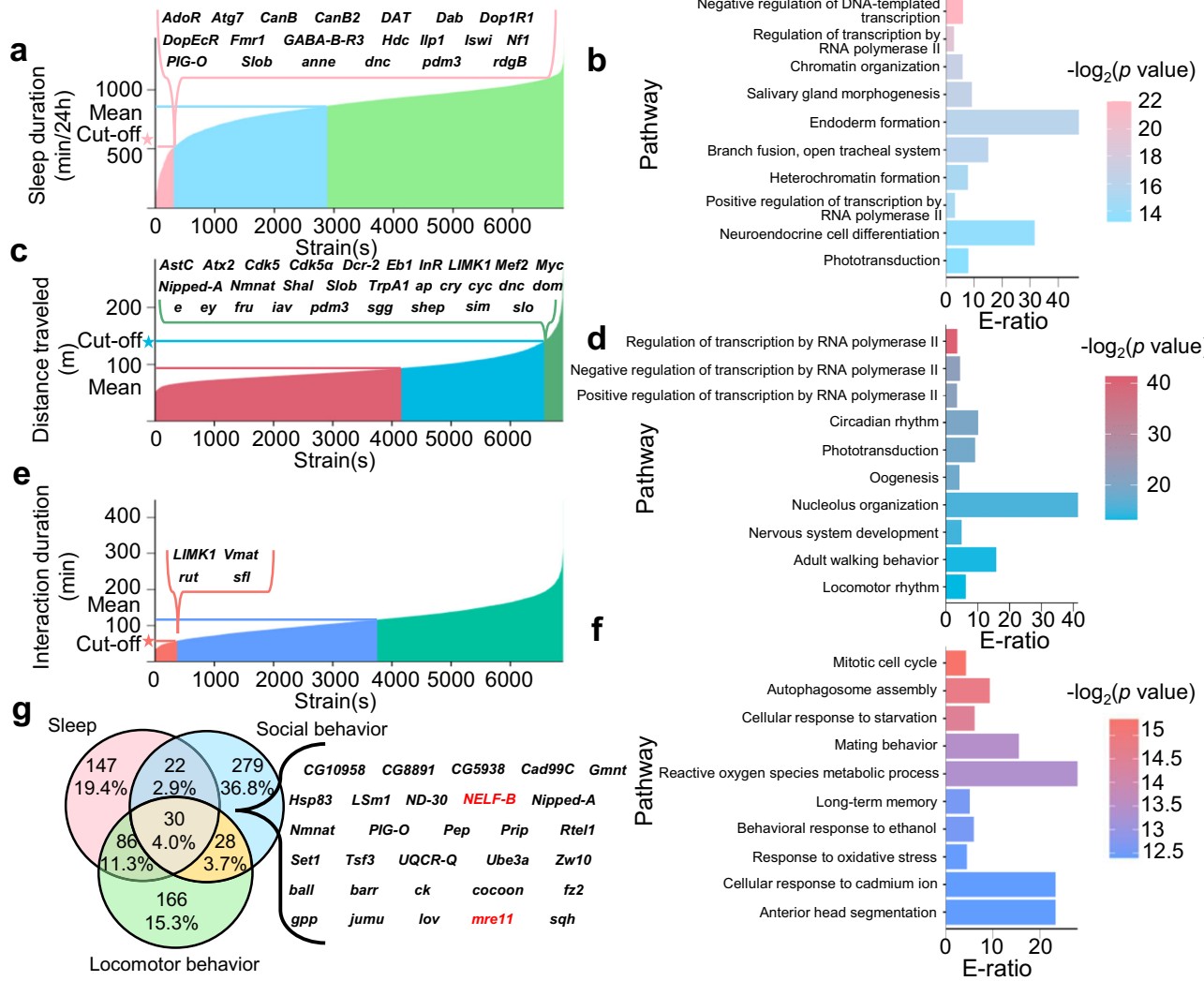

**Fig. 2 | Genome-wide screen of genes involved in regulating sleep, locomotor, and social activity.** Male flies are monitored under 12 h light/12 h dark (12L12D) condition. **a** The daily sleep duration of flies in the screen. Pink indicates fly lines with sleep duration 2 SDs shorter than the mean. Blue indicates fly lines with sleep duration shorter than the mean but not beyond 2 SDs. Green indicates fly lines with sleep duration longer than the mean. The 20 known genes reported to be involved in sleep regulation are indicated. **b** GO enrichment analysis of the 285 fly genes involved in sleep regulation. E-ratio, enrichment ratio (one-sided hypergeometric test). **c** The daily distance traveled of flies in the screen. Red indicates fly lines that traveled less than the mean. Blue indicates fly lines that traveled more than the mean but less than 50% increase. Green indicates fly lines that traveled 50% more than the mean. The 29 known genes found to be involved in regulating locomotor

activity are indicated. **d** GO enrichment analysis of the 359 genes involved in regulating locomotor activity. E-ratio, enrichment ratio (one-sided hypergeometric test). **e** The daily social interaction duration of flies in the screen. Red indicates fly lines with social interaction duration 50% shorter than the mean. Blue indicates fly lines with social interaction duration shorter than the mean but not beyond 50%. Green indicates fly lines with social interaction duration longer than the mean. The 4 known genes reported to be involved in social activity are indicated. **f** GO enrichment analysis of the 310 genes involved in regulating social activity. E-ratio, enrichment ratio (one-sided hypergeometric test). **g** Venn diagram demonstrating the overlap of genes involved in each of the 3 behaviors. Genes involved in regulating all three behaviors are indicated. Source data are provided as a Source Data file.

insulin-like peptide *Ilp3*[42,43] (Fig. 3p; Supplementary Fig. 18c; Supplementary Data 7). These results indicate a possibility that MRE11 may somehow affect *Hk* and/or *Ilp3* to promote sleep.

### *NELF-B* promotes sleep regardless of the presence of conspecifics

Another top hit is an essential component of the negative elongation factor (NELF) complex, NELF-B, which functions to inhibit transcription elongation linked to promoter-proximal pausing[44]. Knocking down *NELF-B* with 3 RNAi lines (generated from 2 independent constructs) all lead to reduced sleep duration both under group and isolated condition (Supplementary Figs. 19 and 20). This is accompanied by shortened sleep bout duration, reduced sleep bout number, and delayed sleep onset, indicative of impaired sleep maintenance and

initiation. These flies also exhibit decreased social interaction duration and elevated level of locomotor activity (Supplementary Fig. 21). To characterize the underlying mechanism, we used RNAi-3 to knock down *NELF-B* in various brain structures and cell types (Supplementary Fig. 22a). For positive candidates, we further validated with the other two RNAi lines (Supplementary Fig. 22b). In the end, we found that *NELF-B* deficiency in cells expressing GABA-B receptor (GABA-B-R1) is sufficient to recapitulate the short-sleep phenotype of pan-neuronal knockdown of *NELF-B*. Consistently, THIP completely inhibits the effect of *NELF-B* deficiency on sleep duration, while CBZ and nipecotic acid (NipA) which enhances GABA signaling partially inhibit the effects of *NELF-B* on sleep (Supplementary Fig. 23)[22,45]. In summary, these results strongly suggest that *NELF-B* acts in GABA-B-R1+ cells and impinge on GABA signaling to promote sleep.

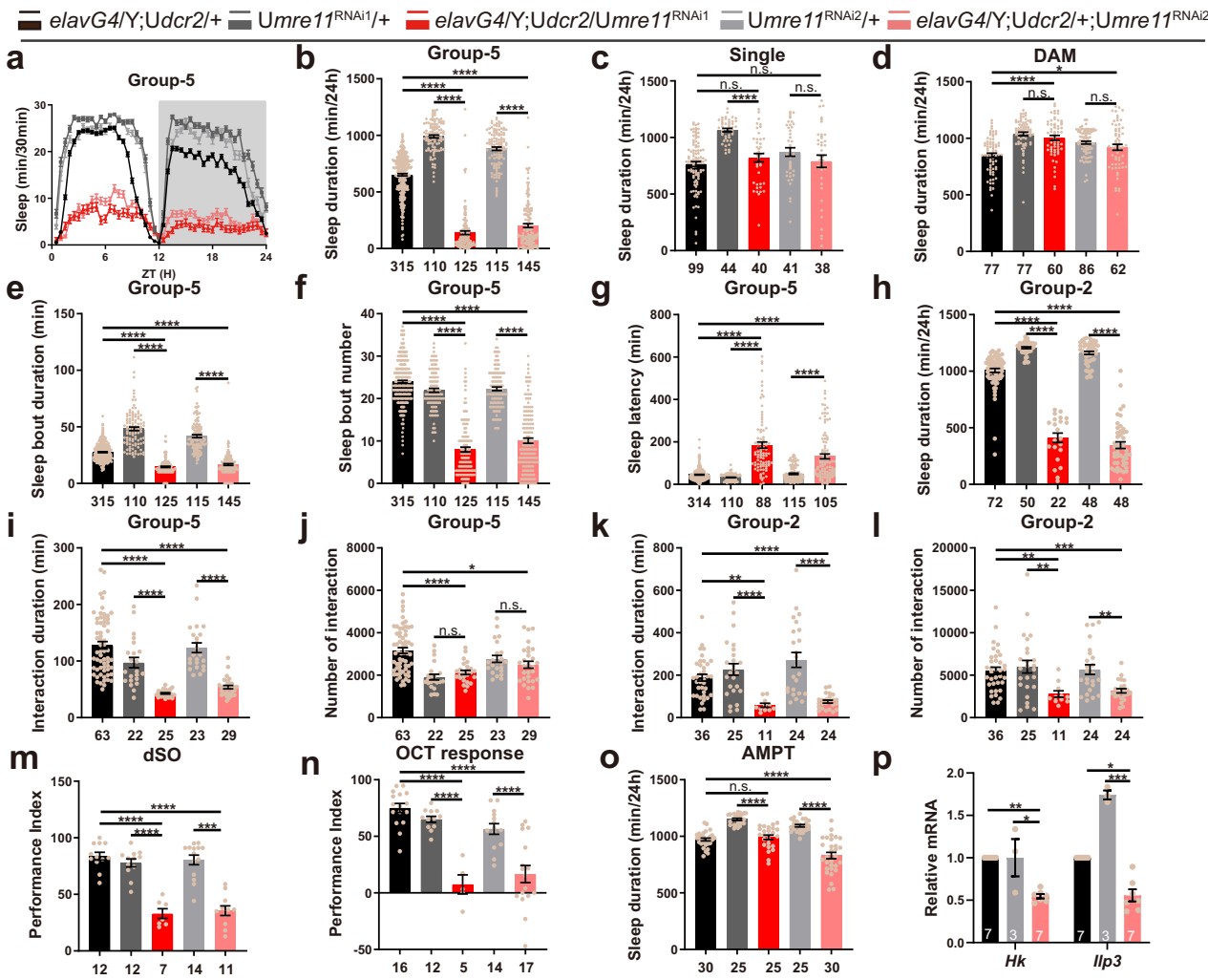

**Fig. 3 | MRE11 functions to promote sleep only in the presence of conspecifics.**
Male flies are maintained under 12L12D condition. **a** Sleep profile of *mre11* RNAi and control flies in group size of 5 (group 5). Gray shade indicates the dark period. The number of flies tested is denoted in (**b**). **b** Daily sleep duration of flies in (**a**). ****$p < 0.0001$. Daily sleep duration of *mre11* RNAi and control flies in isolated condition monitored by video (**c**) or DAM system (**d**). *$p = 0.0205$, ****$p < 0.0001$. Sleep bout duration (**e**), sleep bout number (**f**) and sleep latency (**g**) of *mre11* RNAi and control flies in group 5. ****$p < 0.0001$. **h** Daily sleep duration of *mre11* RNAi and control flies in group size of 2 (group-2). ****$p < 0.0001$. Daily social interaction duration (**i**) and number of social interaction (**j**) (*$p = 0.0104$) of *mre11* RNAi and control flies in group 5. ****$p < 0.0001$. Daily social interaction duration (**k**) (**$p = 0.0377$, ****$p < 0.0001$) and number of social interaction (**l**) (from left to right, **$p = 0.0028$, **$p = 0.0034$, ***$p = 0.0002$, **$p = 0.0013$) of *mre11* RNAi and control flies in group-2. **m** Performance index of *mre11* RNAi and control flies on dSO test

when responder and emitter flies share the same genotype. ***$p = 0.0001$, ****$p < 0.0001$. **n** Performance index of response to OCT. A higher performance index indicates increased avoidance of OCT. ****$p < 0.0001$. **o** Daily sleep duration of *mre11* RNAi and control flies with AMPT. ****$p < 0.0001$. **p** Relative mRNA level of sleep genes in whole heads of *mre11* RNAi and control flies. *Hk* *$p = 0.0126$, **$p = 0.0051$; *Ilp3* *$p = 0.0144$, ***$p = 0.0007$. The number of flies (**b–h**, **o**)/wells (**i–l**)/ independent experiments (**m**, **n**, **p**) tested is denoted on or below each bar. Kruskal–Wallis and Dunn's multiple comparisons test was used in (**b–j**, **l**, **p**, RNAi2 *vs.* UAS/GAL4 controls in **k** and **m**, RNAi1 *vs.* UAS/GAL4 controls in **o**). One-way ANOVA and Sidak's multiple comparisons test were used in (**n**, RNAi1 *vs.* UAS/GAL4 controls in **k** and **m**, RNAi2 *vs.* UAS/GAL4 controls in **o**). Data are presented as the mean ± SEM. n.s. not significant; G4, GAL4; U, UAS. Source data are provided as a Source Data file.

## Construction of a signal web by LLM-reasoning to demonstrate molecular signatures that control sleep, locomotor, and social activities

In order to further excavate the molecular connections between sleep, locomotor, and social behavior that lie within our screen results, we generated an educated signaling network consisting of a total of 86 genes that are identified by the screen. These include the 19 known genes interpreted by GPT-3.5 to be involved in regulating sleep, locomotor, and/or social activity, and another 67 genes that either have at least two independent RNAi lines demonstrating similar phenotypes or are among the top candidates with the strongest phenotypes (Fig. 4a; Supplementary Data 8a). Based on gene functions provided by GPT-3.5, these genes were classified into 7 categories. Interestingly, genes

that belong to calcium and intracellular signaling pathway only participate in sleep and/or social behavior regulation, whereas genes involved in neurotransmission and synaptic function almost exclusively participate in regulating sleep and locomotor activity. Genes in the other 5 categories are involved in all three behaviors.

To reveal potential functional regulations or associations of every member of the network with all the other members, LLMs were used to generate a total of 3655 answers using the CoT prompting strategy (Supplementary Data 8b). From the results, 139 pairs of potential regulations or associations were acquired via reasoning, and 103 (74.1%) of these answers were supported by the literature (Supplementary Data 8c). The direction of information flow is clear for 64 of these pairwise relationships, while the remaining are uncertain, and

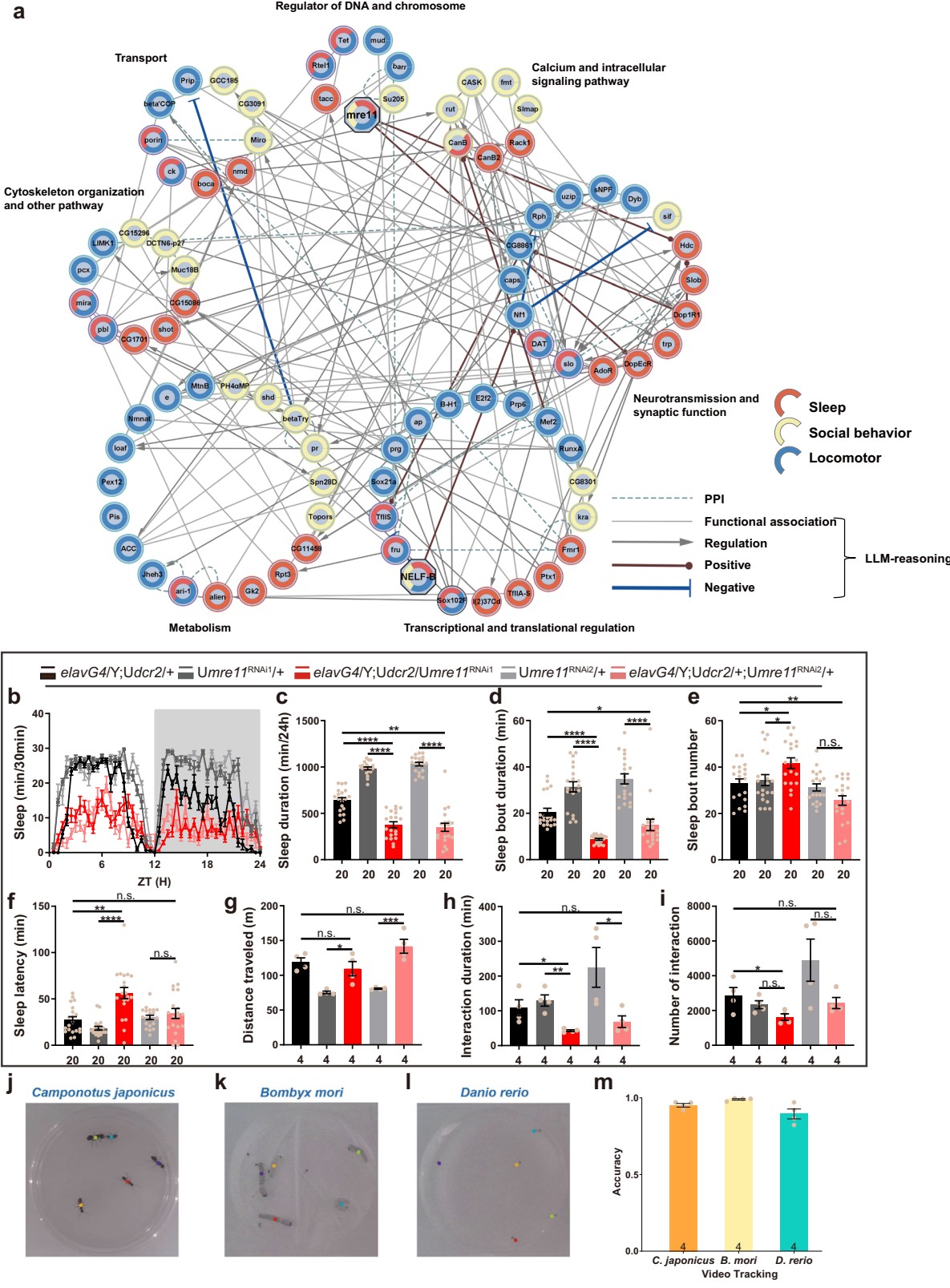

current knowledge suggests that they might be functionally associated with each other, without detailed mechanisms (Fig. 4a; Supplementary Data 8c). In addition, we incorporated protein-protein interaction (PPI) information from BioGRID database and there are only 17 pairs of protein or genetic interactions, demonstrating the superiority of LLM-reasoning in comprehensive modeling of educated signaling networks. Genes that belong to the category of neurotransmission and synaptic

function exhibit the most intense regulations or associations with other members of the network, implicating that these genes play a central role in integrating signals from other pathways and orchestrating the other pathways to control the 3 behaviors.

LLM-reasoning identifies *mre11* to potentially regulate dopamine receptor *DopEcR*, *Dop1R1,* and *Hdc*, while *NELF-B* potentially regulates *Neurofibromin 1* (*Nf1*) and RNA polymerase II elongation factor *TfIIS*. All

**Fig. 4 | Signaling web generated by LLM-reasoning demonstrates the molecular signatures of sleep, locomotor, and social behaviors. a** A signal web consisting of 86 genes identified by our screen to regulate the 3 behaviors. Functional associations, regulations, and protein or genetic interactions are labeled. The direction of information flow is denoted if available, including type of regulation (positive or negative). **b** Sleep profile of *mre11* RNAi and control flies fed with Dop1R1 antagonist SCH23390 (100 uM) in group size of 5 (group 5). Gray shade indicates the dark period. (*n* = 20 flies for each group). Sleep parameters of *mre11* RNAi and control flies fed with Dop1R1 antagonist SCH23390 (100 uM) in group 5, including daily sleep duration (**c**) (**$p$ = 0.0065, ****$p$ < 0.0001), sleep bout duration (**d**) (*$p$ = 0.0276, ****$p$ < 0.0001), sleep bout number (**e**) (RNAi1 flies *vs.* GAL4 controls *$p$ = 0.0331, RNAi1 flies *vs.* UAS controls *$p$ = 0.0392. **$p$ = 0.0089) and sleep latency (**f**) (**$p$ = 0.0025, ****$p$ < 0.0001). Daily social interaction duration (**g**) (*$p$ = 0.0138,

***$p$ = 0.0003), number of social interaction (**h**) (RNAi1 flies *vs.* GAL4 controls *$p$ = 0.0370, **$p$ = 0.0087, RNAi2 flies *vs.* UAS controls *$p$ = 0.0300) and daily distance traveled (**i**) (*$p$ = 0.0442) of *mre11* RNAi and control flies fed with Dop1R1 antagonist SCH23390 (100 μM) in group 5. Video tracking images of *Camponotus japonicus* (**j**), *Bombyx mori* (**k**), and *Danio rerio* (**l**). **m** Measurement of movement tracking accuracy for three types of animals (*n* = 4 biologically independent experiments). The number of flies (**c**–**f**)/wells (**g**–**i**) tested is denoted below each bar. Kruskal–Wallis and Dunn's multiple comparisons test was used in (**d**, **f**) RNAi2 *vs.* UAS/GAL4 controls in (**c**), RNAi1 *vs.* UAS/GAL4 controls in (**e**). One-way ANOVA and Sidak's multiple comparisons test were used in (**g**, **h**, **i**) RNAi1 *vs.* UAS/GAL4 controls in (**c**), RNAi2 *vs.* UAS/GAL4 controls in (**e**). Data are presented as the mean ± SEM. n.s. not significant; G4, GAL4; U, UAS. Source data are provided as a Source Data file.

5 potential regulations were not covered by the BioGRID PPI data. To validate these predictions, we first measured the mRNA level of these genes in *mre11* and *NELF-B* RNAi flies and found *Hdc* to be significantly reduced (Supplementary Fig. 24). Furthermore, we treated *mre11* flies with drugs that target Dop1R1 and DopEcR (Fig. 4b–i; Supplementary Fig. 25). We found that Dop1R1 antagonist SCH23390 rescues the reduced sleep bout number and partially rescues (effective on only one of the RNAi lines) the lengthened sleep latency in *mre11* RNAi flies (Fig. 4e, f). SCH23390 also rescues the increase of locomotor activity and partially rescues the decrease of social interaction caused by *mre11* deficiency (Fig. 4g–i). We further compared the effects of SCH23390 on each genotype (Supplementary Fig. 26). As can be seen, SCH23390 significantly shortens sleep latency in *mre11* RNAi flies but not the controls. These findings suggest that MRE11 may influence sleep, locomotor, and social activity by modulating *Dop1R1* and *Hdc* in an unknown manner.

Taken together, this educated network demonstrates the molecular signatures of sleep, locomotor, and social behavior, with LLM-reasoning revealing the intricate connections among members of the network.

Lastly, we tested whether our system can be used for monitoring the behaviors of other animals. We recorded the movements of ants, silkworms, and zebrafish, and processed the videos using the VTL method (Figs. 4j–l; Supplementary Movie 4–6). The tracking accuracy for these three animals are 91.15%, 98.96%, and 89.53%, respectively (Fig. 4m). This means the platform and methods that we established here can be easily adapted to investigate sleep, locomotor, and social behaviors in a variety of other animals.

## Discussion

Liu et al. reported substantially reduced daily sleep duration and sleep bout length accompanied by increased sleep bout numbers for fly populations containing 50 individuals compared to fly recorded in isolation[6]. Here, we observe sleep fragmentation with as few as 2 flies in a group, while daily sleep duration displays significant shortening when the group size reaches 3 individuals. This demonstrates the prominent influence of social signal-related sensory cues on sleep quantity and quality.

Considering that extensive genome-wide sleep screens have been conducted in flies using the *Drosophila* Activity Monitoring (DAM) system, we are somewhat surprised by that only 20 of the genes identified here have been previously reported to be involved in sleep regulation. We reasoned that this could in part be due to that for most of the genes only one RNAi line was used and we only expressed these RNAis using *elav*GAL4. There may be insufficient knockdown of a sleep-regulating gene in brain regions where it functions, resulting in false negatives. A more intriguing possibility is that the mechanism which controls sleep in the presence of conspecifics is distinct for the most part from that under isolated condition, and thus here many of the hits have not been previously identified, such as *mre11*. On the other hand, some of the genes that regulate sleep under isolated

condition may be dispensable under group condition. Of course, this will require further validation, and if validated this may subvert our current understanding regarding the molecular mechanism of sleep regulation which is based on studies of animals in isolated condition. Although investigating sleep in the presence of conspecifics is complicated by social cues, this is likely more similar to the natural habitat.

Previous studies have demonstrated that prompt engineering is an effective strategy to elicit text content derived from LLMs for accomplishing various tasks[16,17]. Here in this study, a standard prompting strategy was used to conduct a genome-wide interpretation regarding sleep, locomotor, and social activity, and we show the prompt for *Drosophila* protein Dop1R1 as an example (Fig. 5a, b). In this prompt, we queried about the function of Dop1R1, and then questioned whether this protein is involved in regulating sleep. In the response generated by GPT-3.5, Dop1R1 was described to be involved in sleep regulation and activation of Dop1R1 enhances activity and decreases sleep in fruit flies, which is consistent with the published literature[46]. Furthermore, the CoT prompting strategy was utilized to reveal functional regulations or associations among the members of the signal web constructed (Fig. 5a). The basic descriptions of *mre11* and *Dop1R1* were presented, and we queried about the potential relationships between these two genes and their detailed CoT of LLM-reasoning (Fig. 5c). As can be seen in the answer, GPT 3.5 reasoned that *mre11* regulates *Dop1R1* via dopaminergic signaling pathway. We further validated this (at least in part) by demonstrating that Dop1R1 antagonist SCH23390 partially rescues the sleep, locomotor, and social activity phenotypes of *mre11* RNAi flies. Taken together, our findings demonstrate successful applications of prompt engineering to efficiently utilize LLMs in scientific research.

Our work reveals unrivaled capability of GPT-3.5 in information gathering, interpretation, reasoning, and understanding with a satisfactory low false positive rate. After alignment with human perception, GPT-3.5 successfully learnt human-like CoTs, which are highly useful for scientific research beyond prediction which is the function that has been exploited the most for AI. For the 86 genes in the educated signal web, it is almost impossible for researchers to manually check all pairwise interactions of 86 * (86-1)/2 = 3655 combinations. On the other hand, such a tedious task can be accomplished by GPT-3.5 with ease, accompanied by unambiguous CoTs for reasoning all the connections. In this study, we found that answering 3655 questions by GPT-3.5 only took about 3 h, with ~3 s/answer. On the other hand, we manually curated the experimental evidence of 139 pairs of potential functional regulations or associations from the literature for over one week (~10 h/day), with ~30 min/answer. Thus, GPT-3.5 can dramatically promote our research efficiency.

Of note, in this study, we human researchers proposed the hypotheses, designed and performed the experiments and the data analysis, and wrote the manuscript, while GPT-3.5 helped with most of the information gathering and processing (while the accuracy of these actions were manually checked by us human researchers, or validated by follow-up experiments). Our study demonstrates the application of

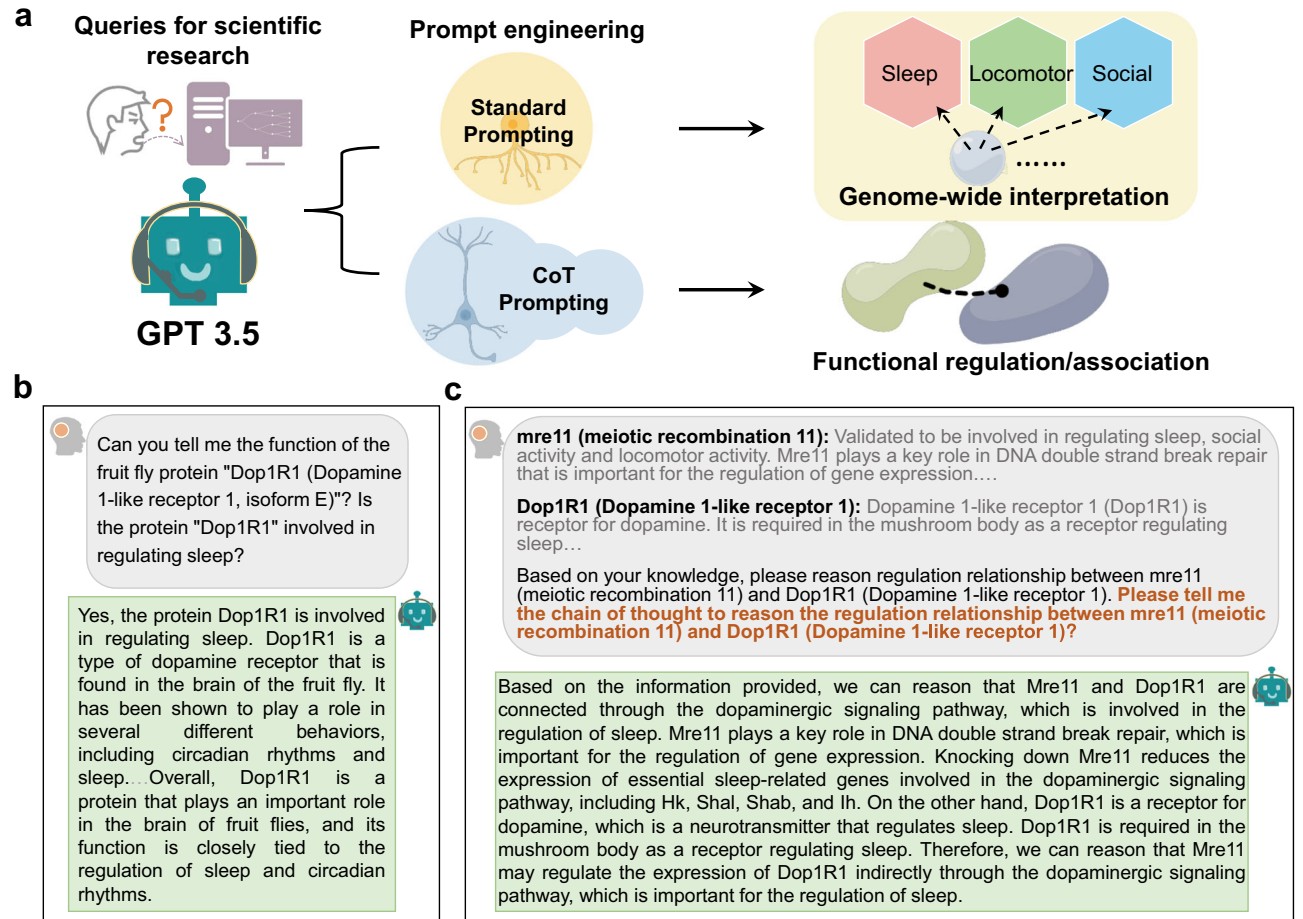

**Fig. 5 | The overview of prompt engineering used in this study. a** Illustration of prompt engineering for eliciting text information generated from GPT-3.5, which includes standard prompting and CoT prompting. The icons of human head, computer, neurons, and molecules are provided by Prof. Jian Ren. **b** An example of a standard prompting that demonstrates the interpretation of a functional role in sleep regulation for Dop1R1. The icon of human brain is provided by Prof. Jian Ren. **c** An example of a CoT prompting that demonstrates regulation of *Dop1R1* by *mre11*. The icon of human brain is provided by Prof. Jian Ren.

LLMs for assisting scientific discovery. We anticipate that tools of this sort shall revolutionize the way we learn knowledge and design scientific research, and look forward to a new era of human-AI interactive scientific practice.

## Methods

### Ethics statement

All studies were approved by the Ethics Committee of Huazhong University of Science and Technology. All zebrafish maintenance procedures and experiments were in accordance with guidelines approved by the Ethics Committee of Huazhong University of Science and Technology (2019S907).

### Fly strains and maintenance

Flies were raised on standard cornmeal food at 25 °C and ~50% humidity under 12L12D. All strains were obtained from Bloomington *Drosophila* Stock Center (BDSC), Vienna *Drosophila* Resource Center, and TsingHua Fly Center or as gifts from colleagues. Fly strains used for genome-wide screen were listed in Supplementary Data 5. Most of the neurotransmitter related GAL4 lines were generated in Dr. Yi Rao's laboratory[47], including *TH*GAL4, *DAT*GAL4, *Dop2R*GAL4, *Dop1R2*GAL4, *SerT*GAL4, *5-HTR1B*GAL4, *TRH*GAL4, *AdoR*GAL4, *VGlut*GAL4, *GluR-IA*GAL4, *ChAT*GAL4, *Tdc2*GAL4, *Dop1R1*GAL4, *Rdl*GAL4, *GABA-B-R1*GAL4, *GABA-B-R2*GAL4 and *GABA-B-R3*GAL4. The *Drosophila simulans w^{501}* strain is a kind gift from Dr. Jian Lu at Peking University. The following fly lines were also used in this study: isogenic *w^{1118}*

(BDSC:5905), *DAT^{fmn25}*, *Pdfr^{han5304}* (BDSC:33068), UAS-*Cul3*RNAi (11861R-2, Fly Stocks of National Institute of Genetics), UAS-*CanA-14F*RNAi (V30105), UAS-*Fmr1* (BDSC:6928), UAS-*NachBac* (BDSC:9466), UAS-*TrpA1* (BDSC:26263), UAS-*mre11*RNAi1 (THU5229), UAS-*mre11*RNAi2 (TH01614.N), UAS-*NELF-B*RNAi1 (THU0696), UAS-*NELF-B*RNAi2 (THU3523), UAS-*NELF-B*RNAi3 (THU4946), UAS-*dcr2* (BDSC: 24650), *elav*GAL4 (BDSC:458), *c305a*GAL4 (BDSC:30829), *c309*GAL4 (BDSC:6906), *30Y*GAL4 (BDSC:30818), *MB247*GAL4 (BDSC:50742), *c547*GAL4[48], *c819*GAL4 (BDSC:30849), *Dilp2*GAL4 (BDSC:37516), *Gad1*GAL4 (BDSC:51630), *VGAT*GAL4 (BDSC:58980), *tim*GAL4 (BDSC:7126), *cry*GAL4-16 (BDSC:24514), *103Y*GAL4 (BDSC:30813), *c42*GAL4 (BDSC:30835), and *Feb170*GAL4[49]. 3–5-day old male flies were used for all experiments except for those indicated otherwise. Flies were euthanized by rapid freezing at −80 °C.

### Maintenance of other animals used

Zebrafish were raised and maintained on a 14L10D cycle at 28.5 °C in a circulating water system. 6 dpf embryo of standard wild-type (AB) laboratory strain in E3 medium (5 mM NaCl, 0.17 mM KCl, 0.33 mM CaCl2, 0.33 mM MgSO4) were used for behavior monitoring. Silkworms and ants were obtained from licensed private pet rearers with no additional information regarding strain and genetic background. The silkworms were fed fresh mulberry leaves and maintained under 12L12D at 25 °C with 70% relative humidity. The ants were kept in a dark plastic box in the incubator (25 °C and 60% relative humidity), and fed with honey and purified water with the weight ratio of 1:2. For behavior

monitoring, first-instar silkworms and 4-month-old ants were used. These animals were only used for testing whether our video-tracking system could be employed for monitoring activity in different species and thus effects of gender, strain, and genetic background were not taken into consideration. Zebrafish, ants, and silkworms were euthanized by rapid freezing at −80 °C.

### Video tracking assay
Video capture and object tracking analysis of fruit flies were performed as below. In brief, five commonly used biochemical incubators were purchased (Ningbo Yanghui Instrument Co., ltd.), and further adapted to accommodate video-tracking instruments (Wuhan Yuda Instrument & Equipment Co., ltd.) which are infrared light camera with a wavelength of 850 nm. The video capture software package, AMcap (https://amcapdl.com/), was used for object tracking in videos with a camera. Male and female flies 3–5 days old were entrained for 1.5–2 days in 12-well plates lined with fly food containing 2% agar, 5% sucrose, and 0.1% propanoic acid, and then their movements were monitored for 24.5 h. Each well is 35 mm in diameter and 5 mm in height and was covered with 0.1 mm thick transparent slide (35 mm × 34 mm) with about 2 mm space between the fly and the cover, which ensures that the flies can move freely. Each incubator contains four levels while each level can accommodate a total of 8 culture plates, allowing 96 wells to be monitored simultaneously. All recordings were conducted at 25 °C under 12L12D except for those stated otherwise.

### The VTL method
After converting the video into frame-by-frame image slices with a media transcoder OpenCV (https://opencv.org), we normalized the size of each well to a circle with a diameter of 430 pixels and then conducted image greyscale and edge correction. In image region segmentation, an area with darker gray color compared with the colors of surroundings were extracted and calculated for determining the contour of a fruit fly. For the analysis of fly location, a 2-dimensional coordinate was laid on the circle representing the well with the $x$ and $y$ coordinates as tangent lines along the edge of the circle, and the origin of the coordinate is located to the upper left corner outside of the circle. In this way, the fly location in each frame was converted into a specific coordinate for further calculation and analysis. The resulting file was saved as a.vtl file.

The tracking method for monitoring the movements of ants, silkworm, and zebrafish were performed similarly. Meanwhile, the detailed parameters for reducing image noise and the threshold of contour size were adjusted according to the size and color of different species.

### Definition of the 3 behaviors
**Definition and parameters of sleep.** If the distance between two consecutive frames of each single fly is less than 3 pixels, it is regarded to be in the rest state. When the rest state of a fly lasts for longer than 5 min[50,51], this fly is considered to be in the sleep state. Based on this criteria, daily sleep duration, sleep bout length, sleep bout number, and sleep latency can be calculated.

**Definition and parameters of locomotor activity.** The locomotor activity of the fruit flies was detected as movements and average daily distance traveled for each fly in a well was calculated[52].

**Definition and parameters of social activity.** The distance between 2 fruit flies were calculated that range from 5 pixels to 100 pixels, and when the distance between 2 flies is within 40 pixels, this is defined as a social interaction. The social behavior for each time was recorded when the distance between 2 flies was less than 40 pixels until their distance was beyond 40 pixels. In this way the duration and frequency of social interactions can be calculated. Data presented are the average daily social interaction duration and frequency of each fly pair in a well.

### Mapping of fly protein isoforms across the genome
A total of 31,748 fly gene entries were identified in the FlyBase (https://flybase.org/)[53]. Considering the presence of protein isoforms, the full name of proteins were mapped and retrieved from UniProt (https://www.uniprot.org/)[54]. By curating these information regarding the proteins and their isoforms, a total of 42,794 protein isoforms across the entire fly genome were acquired for further analysis.

### The prompting strategy
For prompt engineering, there were two rounds in querying GPT-3.5. In the first round, GPT-3.5 was queried with the initial questions, and then we manually checked the confidence of each answer. Ambiguous answers with less confidence were discarded. If the answer was affirmative without any ambiguity, a second round was carried out. We further included informative hints when querying GPT-3.5 as a priming knowledge, and then queried the machine CoT that reasoned the answer. By eliciting the LLMs to produce responses with CoTs, GPT-3.5 was elicited to generate high-quality answers in a boosting-like pattern. A typical example of the two-round prompting strategy is shown below:

(i) The first round.

Prompt 1 (by human researchers):

"Can you tell me the function of the fruit fly protein "Dop1R1 (Dopamine 1-like receptor 1, isoform E)"? Is the protein "Dop1R1" involved in regulating sleep?"

Response 1 (by GPT-3.5):

Yes, the protein Dop1R1 is involved in regulating sleep in fruit flies. Dop1R1 is a type of dopamine receptor that is found in the brain of the fruit fly. It has been shown to play a role in several different behaviors, including circadian rhythms and sleep. In one study, it was found that mutations in the gene encoding Dop1R1 disrupted the sleep patterns of fruit flies. These flies had decreased sleep and altered sleep patterns compared to normal flies. This suggests that Dop1R1 is involved in the regulation of sleep in fruit flies. Overall, Dop1R1 is a protein that plays an important role in the brain of fruit flies, and its function is closely tied to the regulation of sleep and circadian rhythms.

(ii) The second round.

Prompt 2 (by human researchers):

We have known that fruit fly protein Dop1R1 (Dopamine 1-like receptor 1, isoform E) is involved in regulating the sleep of fruit flies. Why do you consider this protein Dop1R1 (Dopamine 1-like receptor 1, isoform E) involved in regulating sleep? Please tell me the chain of thought one by one.

According to the chain of thought you mentioned, which experiments must be performed to validate this function in sleep regulation? Please list the experimental plan one by one.

Response 2 (by GPT-3.5):

Dopamine has been previously implicated in regulating sleep in fruit flies. 2. Dop1R1 is a receptor for dopamine in fruit flies. 3. Studies have shown that mutations in the Dop1R1 gene can affect sleep patterns in fruit flies. 4. Manipulating the expression or activity of Dop1R1 in fruit flies can also alter their sleep patterns. 5. Further studies have confirmed the involvement of the Dop1R1 protein in sleep regulation through neurotransmitter signaling.

Experimental plan to validate the function of Dop1R1 in sleep regulation: 1. Use genetic manipulation techniques, such as CRISPR/Cas9, to create mutations in the Dop1R1 gene in fruit flies. 2. Analyze the sleep patterns of the mutated fruit flies and compare them to normal fruit flies. 3. Use pharmacological agents to activate or block Dop1R1 activity in fruit flies and observe the effects on their sleep patterns. 4. Conduct electrophysiological recordings to measure changes in neural activity when Dop1R1 is activated or blocked in fruit flies. 5. Use optogenetic techniques to manipulate the activity of dopamine neurons that project to Dop1R1-containing neurons in the brain and measure the resulting changes in sleep patterns. 6. Conduct

behavioral assays to determine whether changing the activity of Dop1R1 affects other behaviors besides sleep.

## Evaluation of the positive response rate

To estimate the positive response rate of prompt-generation pairs from LLMs, we defined a list of 27 key phrases including "am not able", "am unable", "cannot access", "cannot conclude", "cannot confirm", "cannot find", "cannot provide", "cannot retrieve", "cannot say", "cannot state", "cannot tell", "couldn't find", "do not currently have", "do not have", "do not know", "don't have", "don't know", "has not been determined", "no documented evidence", "no evidence", "no known connection", "no known direct association", "no published research", "no scientific evidence to", "not able to provide", "not clear", and "not found to be directly involved". Answers without any one of the key phrases mentioned above were considered as positive responses.

## Benchmark data preparation

To evaluate the performance of GPT-3.5 on genome-wide interpretation, we conducted a literature curation to collect experimentally identified genes known to be involved in regulating sleep, locomotor, and social activity from PubMed, using multiple keyword combinations, such as "Drosophila and sleep", "Drosophila and locomotor" and "Drosophila and social gene", respectively. In total, 268, 283, and 49 fly genes were obtained to be reported in regulating sleep, locomotor, and social activity, respectively (Supplementary Data 2). For each reported gene, primary references were provided with corresponding PMIDs.

For each behavior, the known regulatory genes were taken as positive data to calculate Type II errors. To unbiasedly estimate Type I errors, we randomly selected an equal number of genes from genes that have not been reported to be involved as negative data. Such a resampling procedure was repeatedly performed for 20 times, and then the average Sp value was calculated for each behavior. To ensure the reproducibility of this study, both positive and negative datasets were presented, and individual gene information including gene symbol, gene name, and accession numbers were integrated from the FlyBase database (Supplementary Data 2).

## Performance evaluation

By comparison of the benchmark datasets with GPT-3.5 interpretations, true positive (TP), true negative (TN), false positive (FP), and false negative (FN) values were separately calculated for the interpretation of individual gene regarding sleep, locomotor, and social activity (Supplementary Data 2). Then, 5 measurements, including Sn, Sp, accuracy (Ac), Mathew correlation coefficient (MCC), and precision (Pr), were calculated as follows:

$$\mathrm{Sn} = \frac{\mathrm{TP}}{\mathrm{TP} + \mathrm{FN}} \tag{1}$$

$$\mathrm{Sp} = \frac{\mathrm{TN}}{\mathrm{TN} + \mathrm{FP}} \tag{2}$$

$$\mathrm{Ac} = \frac{\mathrm{TP} + \mathrm{TN}}{\mathrm{TP} + \mathrm{FP} + \mathrm{TN} + \mathrm{FN}} \tag{3}$$

$$\mathrm{MCC} = \frac{(\mathrm{TP} \times \mathrm{TN}) - (\mathrm{FN} \times \mathrm{FP})}{\sqrt{(\mathrm{TP} + \mathrm{FN}) \times (\mathrm{TN} + \mathrm{FP}) \times (\mathrm{TP} + \mathrm{FP}) \times (\mathrm{TN} + \mathrm{FN})}} \tag{4}$$

$$\mathrm{Pr} = \frac{\mathrm{TP}}{\mathrm{TP} + \mathrm{FP}} \tag{5}$$

For each behavior, the Sn value was directly calculated, whereas the average values of Sp and other measurements were obtained from 20 times of resampling, with a ratio of 1:1 between the positive and negative datasets. Confusion matrices were separately illustrated for assessing the performance of LLM regarding sleep, locomotor, and social activities.

## GO Enrichment analysis

For enrichment analysis of the differentially regulated proteins participating in sleep, locomotor, and social activity, the GO annotation file (released on March 6, 2023) was downloaded from the Gene Ontology Consortium (http://www.geneontology.org/)[55], and 12,745 fruit fly genes with at least one annotated GO term were obtained. For each GO term $g$, we defined the following:

$N$ = the number of total proteins annotated by at least one GO term,

$n$ = the number of total proteins annotated by GO term $g$,

$M$ = the number of differentially regulated proteins annotated by at least one GO term, and

$m$ = the number of differentially regulated proteins annotated by GO term $g$.

Then, the enrichment ratio (E-ratio) of GO term $g$ was calculated, and the $p$-value was calculated as shown below using hypergeometric distribution:

$$\mathrm{E} - \mathrm{ratio} = \frac{m}{M} \Big/ \frac{n}{N} \tag{6}$$

$$p\,\mathrm{value} = \sum_{m'=m}^{n} \frac{\binom{M}{m'}\binom{N-M}{n-m'}}{\binom{N}{n}} (\mathrm{E} - \mathrm{ratio} \geq 1) \tag{7}$$

## Infrared-based acquisition systems and behavioral analysis

*Drosophila* Activity Monitor system (Trikinetics) was used to analyze fly sleep by recording infrared beam breaks. Male flies 3-5 days old were used for experiments. Flies were entrained under 12L12D at 25 °C for 3 days, and then their activities in the next 4 days under 12L12D were monitored. Sleep was analyzed using SleepMat[56].

## Sleep deprivation assay

Mechanical sleep deprivation was performed at 25 °C using a multi-tube vortexer (VWR) modified by TriKinetics to house DAM2 activity monitors. After 3 days of 12L12D entrainment and 1 day of baseline sleep recording, the multi-tube vortexer delivered 10 s-long vibrations at random intervals centered around 1 min. The intensity of the vortexer was set to 4. The sleep loss percentage was calculated as the amount of sleep loss divided by baseline sleep duration.

## Thermogenetic neuron activation

UAS-*TrpA1* (BDSC:26263) was used for the thermogenetic activation of neurons. Before the behavioral test, flies were raised at 20 °C, and baseline sleep was monitored at 20 °C. Temperature was then raised at lights on to 28 °C for further behavioral monitoring.

## Drug treatment

Drugs were mixed into fly food at the indicated concentrations and the flies were fed with these food for the entire monitoring period. Concentrations of different drugs are as follows: 2 mg/ml AMPT (Sigma, 658-48-0), 5 mg/ml L-Dopa (BBI Life Sciences, 59-92-7), 2 mg/ml dopamine hydrochloride (Sigma, 62-31-7), 5 mg/ml octopamine hydrochloride (Sigma, 770-05-8), 10 mg/ml NipA (Sigma, 498-95-3), 5 mg/ml ethanolamine O-sulfate (EOS) (Sigma, 926-39-6), 0.2 mg/ml CBZ (Ourchem, 298-46-4), 10 µg/ml THIP (Sigma, 85118-33-8), 100 µM SCH23390 (MedChemExpress, HY-19545A), 100 µM SKF38393 (MedChemExpress, HY-12520A), and 100 µM 20E (MedChemExpress, HY-N0179).

### RNA extraction and quantitative real-time PCR (qRT-PCR)

Approximately 30 5-day-old flies were collected in 15 ml tubes and frozen immediately on dry ice. After vortexing the tube, fly heads were isolated and homogenized in Trizol reagent (Invitrogen, 15596018CN). Total RNA was extracted according to the manufacturer's instructions and qRT-PCR was conducted following our previously published procedures[57]. In brief, cDNA was prepared using TransScript One-Step gDNA Removal and cDNA Synthesis SuperMix (Transgen Biotech, AT311-03). Then qPCR was performed on StepOne Real-Time PCR Systems (Thermo Fisher Scientific) using TransStart Tip Green qPCR SuperMix (TransGene Biotech, AQ141-04). The comparative $\Delta\Delta CT$ method was used to quantify the relative gene expression using *rp49* as an endogenous control. The sequences of primers used are in Supplementary Data 7.

### *Drosophila* stress odorant (dSO) avoidance assay

The dSO avoidance assay was performed using a T-maze[58]. After 30 min of acclimatization, 50 3-5-days-old male responder flies were transferred into the elevator of the T-maze apparatus. And 40 3-5-days-old male emitter flies were vortexed for 55 s (15 s on and 5 s rest, repeated 3 times) and subsequently removed from the vial. This vial filled with dSO signal was placed on the T-maze along with a fresh vial containing ambient air. Then responder flies were allowed to choose between the fresh vial and the dSO vial for a 1-min period followed by counting the number of flies in each vial. The performance index was calculated by subtracting the number of flies in the dSO vial from those in the fresh vial, and this difference is then divided by the total number of flies in the two vials.

### Social space assay

The social space assay was performed based on a previous study[59]. In short, 40 3-5-days-old male flies were acclimated to the behavioral room condition for 2 h prior to being loaded into a vertical arena via aspiration. Flies were allowed to explore freely in the triangular chamber. Once flies settled (~30 min after flies were placed in the chamber), pictures of each chamber were taken and the location of each fly was recorded. Information of the distance to the closest neighbor was analyzed by ImageJ.

### OCT response assay

After 30 min of acclimatization, 50 3-5-days-old male flies were transferred into the elevator of the T-maze apparatus and were allowed a 2 min period to choose between fresh air and 0.15% OCT (Sigma, 589-98-0). A preference index was calculated by subtracting the number of flies on the air side by the number of flies on the stimulus side and this difference is then divided by the total number of flies.

### Electric shock response assay

After 30 min of acclimatization, 50 3-5-days-old male flies were transferred into the elevator of the T-maze apparatus and were allowed a 2 min period to choose between two vials with or without electric shock (90 V). A performance index was calculated by subtracting the number of flies on the control side by the number of flies on the shock side and this difference is then divided by the total number of flies.

### LLM-reasoning

For comprehensive understanding of potential functional regulations or associations between each pair of genes in the molecular network, GPT-3.5 was used to generate a total of 3655 answers using the CoT strategy. Then, we defined a list of 11 key phrases including "can reason", "could reason", "can be reasoned", "could be reasoned", "reasonable", "can be hypothesized", "can hypothesize", "can be inferred", "can infer", "can reasonably assume", and "plausible to hypothesize". Answers which contain anyone of the above key phrase and do not

contain the negative keyword "no" are considered as positive responses (Supplementary Data 8).

For the 139 prompt-generated positive responses, gene-gene relationships without unambiguous regulatory directions were defined as "Functional associations" (gene *A* potentially regulates gene *B* or vice versa). Other relationships with the direction of information flow were defined as "Regulations" (gene *A* regulates gene *B*). Also, the type of regulation, positive or negative, was indicated if gene *A* positively or negatively regulates gene *B*, respectively. For each response, we carefully search PubMed to find literature evidence, and primary references were provided with PMIDs if available (Supplementary Data 8).

### Construction of the educated signal web regulating the 3 behaviors

Among the genes identified in our screen to be involved in regulating sleep, locomotor, and social activity, there are 19 genes interpreted by GPT-3.5 to be involved and 6 genes with at least two independent RNAi lines demonstrating similar phenotypes. These genes were selected for modeling the signal web. In addition, 22 genes that when knocked down result in a sleep duration 4 SDs shorter than that of the population mean were selected. 19 genes that when knocked down result in a social interaction duration that is shorter or equal to 1/3 of the population mean were also selected, as well as 29 genes that when knocked down result in at least a 2-fold increase of locomotor activity. In total, 86 non-redundant genes were curated to depict the underlying molecular signatures controlling the 3 behaviors. Combining LLM-reasoning with manual curation, these genes were classified into 7 categories, including regulator of DNA and chromosome, transcriptional and translational regulation, neurotransmission and synaptic function, calcium and intracellular signaling pathway, metabolism, transport, as well as cytoskeleton organization and other pathway. Pre-compiled PPIs were retrieved from a public database, BioGRID (https://thebiogrid.org)[60].

### Statistical analysis

For experiments in this study, quantifications in all experimental graphs represent the mean of at least three biological replicates, and error bars represent the standard error of the mean (SEM). Before statistical analyses, the Shapiro−Wilk test was used for assessing the normal distribution of the data. Welch's *t*-test or the Mann-Whitney test was used for comparison between two independent groups. One-way ANOVA followed by Sidak's multiple comparisons test or the non-parametric Kruskal−Wallis test followed by Dunn's post hoc test correction was used for multiple comparisons. Two-way ANOVA followed by Tukey's or Holm-Sidak's multiple comparisons test were used for for two categorical independent variables. In each statistical analysis, number of biological replicates and significant *p*-values were noted in the figure legends. Asterisks reflect the calculated *p*-values that are less than 0.05. ns, $p > 0.05$; $^{*}p < 0.05$; $^{**}p < 0.01$; $^{***}p < 0.001$; $^{****}p < 0.0001$.

### Reporting summary

Further information on research design is available in the Nature Portfolio Reporting Summary linked to this article.

## Data availability

The processed video-tracking data are available at https://vtl.biocuckoo.cn/download.html. The data of fly gene entries are downloaded from FlyBase [https://ftp.flybase.net/releases/FB2023_01/precomputed_files/genes/gene_map_table_fb_2023_01.tsv.gz][53], and the protein names are retrieved from UniProt [https://www.uniprot.org/uniprotkb?query=(taxonomy_id:7227)][54]. The data of GO annotation file (released on March 6, 2023) is obtained from the Gene Ontology Consortium [https://ftp.ebi.ac.uk/pub/databases/GO/goa/old/FLY/goa_fly.gaf.116.gz][55]. The PPI data are retrieved from BioGRID

[https://downloads.thebiogrid.org/File/BioGRID/Release-Archive/BIOGRID-4.4.202/BIOGRID-ALL-4.4.202.mitab.zip][60]. The response data produced by GPT-3.5 and other remaining experimental data generated in this study are provided in the Supplementary Information/Source Data file. Source data are provided with this paper.

## Code availability

The source codes of VTL and others used in this study have been deposited to GitHub (https://github.com/BioCUCKOO/VTL) with the DOI identifier (https://doi.org/10.5281/zenodo.10681859)[61].

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

## Acknowledgements

The authors thank Dr. Wanshan Ning for his significant efforts in helping to draft the design of the video-tracking instrument. We would like to thank Drs. Jian Lu, Yi Rao, Liming Wang, Chang Liu, Yufeng Pan, Junhai Han, Magaret Ho, and Yi Zhong for kindly providing flies used in this study. We would like to thank Dr. Mugen Liu for providing zebrafish used in this study. We would like to thank Prof. Jian Ren for providing 11 types of icons in Figs. 1a, f, and 5a–c, which can be published under a CC-BY license. We would like to thank Prof. Bin Liu and his student, Jiangyi Shao, for helpful suggestions regarding LLM-related descriptions. This work is supported by grants from the Natural Science Foundation of China (32341021, 31930021, and 32022035 to L.Z., 32341020 and 31970633 to Y.X, and 81701567 to D.P.), the Ministry of Science and Technology of China (STI 2030-Major Projects 2021ZD0203200-02 to L.Z. and 2022YFC2704304 and 2021YFF0702000 to Y.X.), Hubei Innovation Group Project (2021CFA005 to Y.X.), Hubei Provincial Fund for Distinguished Young Scholar (2022CFA049 to L.Z.) and China Postdoctoral Science Foundation (2019T120648 to D.P.).

## Author contributions

L.Z. and Y.X. conceived the project and oversaw all aspects of the project. D.P., D.L. and C.H. performed the LLM-based interpretation and reasoning. Y.X., L.Z., and L.Z. participated in designing the video-tracking instrument. D.L. wrote the computational code for video data analysis. L.Z. conducted the genome-wide screen and further experimental validations with the help of X.W., Y.Y., L.S., Y.W., J.L. and X.Y., D.L., L.Z., D.P. and C.H. performed data analysis with the help of Yuxiang W., M.Z., Z.F., X.H., M.Ch. and Y.G., L.Z., D.P. and Y.X. wrote the manuscript with input from all the authors. All authors reviewed and approved the manuscript for publication.

## Competing interests

The authors declare no competing interests.
