## [Peer Review File · Nature Communications]

Large-language models facilitate discovery of the molecular signatures regulating sleep and activityREVIEWER COMMENTS

Reviewer #1 (Remarks to the Author):

This is an interesting set of studies that will be influential for shaping the direction of sleep research. The use of ChatGPT is, in some ways, a gimmick. However, the authors used a rationale strategy to identify genes and I found this approach to be refreshing. The sooner the field begins using these tools the better. I applaud them for taking this approach and believe that others will follow. I am also excited by evaluating sleep in individuals and in groups. There is a growing awareness that sleep is more plastic than previously thought. These types of studies have an opportunity to identify critical genes that may have been missed in previous screens. What these studies will tell us about sleep function is unclear as the approach is very new. Nonetheless, the approach will open the way for a better understanding of sleep function and regulation.

I do not have any major critiques. The paper is well written. The authors include all of the appropriate genetic controls. Moreover, the authors conduct a few studies to evaluate how a gene might alter pathways that could indirectly impact sleep (e.g. alternative interpretations (e.g. stress odors). Importantly, the authors interpretation of the data was measured and was consistent with the data presented.

Reviewer #2 (Remarks to the Author):

Review Comment on "Large-language models completely reveal the molecular signatures regulating sleep and activity"

Overall, the authors present an impressive study that utilizes a Large Language Model (LLM) to summarize relevant information and investigates the genetic basis of sleep, locomotor activity, and social behavior in flies. The development of a multiple fly video-tracing system and the large-scale behavior screen add valuable insights to the field. The finding that *mre11* regulates sleep specifically in a group condition is particularly noteworthy. However, there are some issues and areas that need further clarification.

In the section on "Genome-wide LLM interpretation of the genetic basis of sleep, locomotor activity", for example, the authors mention that 12.5% of 42,794 fly protein isoforms were interpreted in sleep regulation (Fig. 1b and Supplementary Table 1). However, concerns arise regarding the accuracy and completeness of this interpretation, as only approximately 30% of the genes reported to be involved in sleep regulation, based on our incomplete collection, are found in the authors' list (Table S2A). To support the claims made in Fig 1c (also, line 86), it is important to provide references for the manually verified genes listed in Table S2A. Additionally, it is crucial to clarify the exact number of genes with positive responses and the accuracy of these responses. Since this is the first time the model has been used for predictions, more precise validations are necessary. Simply categorizing responses as "positive" without further detail is insufficient. Providing more comprehensive judgments would facilitate a better assessment of the method's effectiveness. Additionally, the accuracy of ChatGPT interpretation ranges from 78.3%-82.2% seems not accurate enough for some mature fields. The authors should mention the inadequacy of the ChatGPT, not only to exaggerate its advantages.

In the section on "Development of multiple fly video-tracing instrument", it would be beneficial to include more details about group fly sleep profiles, such as sleep curves, bout duration, bout number, and sleep latency in Extended Data Fig1a. Similarly, including sleep curves, bout duration, bout number, and sleep latency for the positive controls in Extended Data Fig2 would provide a better understanding of the results. Activity plot of *pdfR* mutants are required as its typical pattern in

morning and evening anticipation. Furthermore, it is important to compare these profiles with previous studies to assess their consistency.

The supplementary videos showed error examples of video-to-location (VTL) strategy. Does the error percentage rise with the increasing number of flies?

Regarding the comment about increased sleep and reduced locomotor activity often being caused by sickness, it is necessary to provide references supporting this claim (line 152-153). Given the different screening methods employed, particularly the group tracing approach, it is understandable that newly identified genes involved in sleep may differ from those reported in previous studies. However, it is surprising that only nine genes overlap between the 300+ genes identified in this study and the known genes (line 155-157). To address this, the authors should consider selecting more previously reported genes involved in sleep regulation as methodological positive controls or manipulating the neurons involved in sleep regulation to refine their tracking or experimental methods.

In the section on "mre11 and NELF-B regulate Drosophila sleep", clarification is needed regarding the interaction frequency for mre11 flies (Fig3i and j). Considering the possibility of a higher interaction number rather than duration disrupting fly sleep, this aspect should be addressed.

The authors propose that olfactory cues facilitate sleep when conspecifics are present and impaired sensory processing triggers sleep loss selectively under the group condition (line 210-212). However, this proposal appears inconsistent with the results presented in the study. Therefore, it is recommended to approach the interpretation of the results more cautiously based on the following points: (i) the significant reduction in sleep duration when five or more flies are grouped together suggests that olfactory cues do not promote sleep, at least in this context; (ii) previous studies have indicated that socially-enriched flies sleep more than socially-impooverished flies, suggesting that early social interaction might promote sleep (reference 4, activity tracing is isolated); (iii) for mre11 RNAi flies, the mutant shows defects in olfactory response and group sleep only, which is inconsistent with the provided reference. If the reference is valid, mre11 mutant flies should exhibit reduced sleep even under single tracing conditions, which is not observed. Therefore, a more reasonable interpretation could be that knockdown of mre11 enhances some social interaction-related signals, strongly inhibiting sleep, rather than being a defect but a gain of function. This would explain why isolated tracing does not result in sleep reduction.

In Extended Data Fig.10, while the expression of some sleep-promoting genes is reduced, it is important to investigate whether the expression of sleep-inhibitory genes changed in these flies. Interpreting these results comprehensively would provide a more complete understanding of the molecular mechanisms underlying sleep regulation. Also, the authors should give more control and experiment group, such as UAS control or another RNAi line, to convince this result.

In line 250-253 (and Extended fig15a), the observation that Dop2R-GAL4>UNELF-B-RNAi shows the largest reduction in sleep. It would be beneficial to include additional experimental data or analysis to validate function of UNELF-B in Dop2R neurons, as in Extended Data Fig 15b.

In the section on "Construction of a signal web via complete LLM-reasoning to demonstrate molecular signatures".

Firstly, it appears that the 139 potential interactions depicted in Figure 4 do not overlap with the 17 protein interactions among documented in the BioGRID database. If these predictions are valid, it is crucial to explain why there is no correspondence between the predicted interactions and the protein interaction information in the BioGRID database. Clarifying this disparity would help establish the reliability and uniqueness of your findings.

Secondly, it would strengthen the effectiveness of their predictions if the authors could provide

validation methods. For instance, validating interactions that influence gene expression, such as *mre11*'s regulation of *dop1R1* expression (as mentioned in Fig4a and Table S7C, despite inconsistency with Extended Data Fig10b), or employing alternative means to verify the accuracy of your predictions, would provide more confidence in the proposed molecular interactions. Including additional experimental evidence or validation techniques would be highly beneficial in bolstering the robustness of your conclusions.

Lastly, the suggestion in lines 290-293, where LLM-reasoning is used to potentially support the regulation of dopamine receptors *DopEcR* and *Dop1R1* by *mre11*, which is consistent with previous pharmacological experiments, raises concerns. LLM-reasoning results cannot be further utilized to support the reliability of pharmacological results because the pharmacological results was used as the prompt information in the LLM-reasoning process.

Reviewer #3 (Remarks to the Author):

In this article the authors investigate the genetic regulation of sleep, locomotor and social activity in *Drosophila*. They do this by (1) using object tracking in videos to record the behavior of fruit flies as two-dimensional coordinates of individual flies in a culture plate while knocking down 5,588 genes, and (2) querying the ChatGPT large language model (LLM) whether fly protein isoforms are functionally involved in various behaviors (alone and in conjunction with each other). For the object tracking system, they used an out-of-the box object tracking model, manually evaluating 21,600 frames and reporting 98.58% accuracy. They performed 128,382 queries total, one for each of three behaviors (sleep, locomotor, and social?) * 42,794 fly protein isoforms to elicit predictions from the model as to the involvement of each protein in each behavior. They manually verified responses for known behaviors in the literature, estimating the model's accuracy at 78.3-82.2%. They also explore the *mre11* gene in more detail, which has not been previously reported to participate in sleep regulation.

Since my background/expertise is in machine learning and LLMs, my review will focus on those aspects of the article; I leave validation of the methodological soundness and novelty of the biology to other reviewers with more relevant expertise in that area.

The authors claim to propose a new prompting strategy for eliciting information from the LLM, which they call prime-boost prompt (PBP), to incorporate relevant background knowledge step-by-step into the prompts. The prompting strategy is not clearly described in the article, but the exact prompts used were shared in supplementary material, so I was able to look at them to try to discern more detail on their strategy. In both sets of prompting experiments, their strategy for eliciting information from the model is quite standard, applying known prompting templates to their specific domain of interest, and I do not think it warrants a special name ("PBP"). So, regarding novelty, the question is not whether the LLM prompting strategy is novel, but whether the use of the LLM as a tool enables novel analysis.

The results of the first set of experiments prompting chatGPT are, in my opinion, not very interesting. In this set of experiments the authors ask the model whether a large set of fly proteins are involved in sleep and social activity. As one would expect, the model reflects a fairly accurate knowledge of what is known about these proteins from the scientific literature up until 2021 (the time cutoff for its training data.) The key takeaway here, which is well known, is that LLMs provide a human friendly interface for querying knowledge contained on the web. Pros of such an approach are that the output from a prompt (replacing a search query) is stated in natural english, essentially "Yes, ..." "No, .." or "I don't know..", compared to search results from current search engines such as Google, which, for these types of queries, still respond with a list of relevant websites (in this case, mostly relevant research articles.) Flaws are that we can't know for certain the veracity of the LM text, as current LLM technologies can't accurately attribute their responses to sources/citations. So, this approach is useful for cases where such mistakes are ok. Knowing the relative rates of true positive / false positive / true

negative / false negative predictions would be useful for making informed decisions about appropriate use cases for this approach. Second, largely unlike current search engines, a limitation of current LLM technologies is that the responses are limited to a fixed time frame, based on when data was aggregated for LLM model training. In the case of ChatGPT, this is currently sometime during 2021, though eventually the model will be updated, and consequently get stale (without substantial advances in the area of continual/lifelong learning for LLMs.) On the other hand, new pages and articles can be quickly and easily indexed by current search systems nearly in real time. Third, there are concerns about the replicability of the research, since the closed ChatGPT model was used, which is known to produce different results given the same prompt, due to the stochastic nature of model inference, and the underlying model changes over time, with no access to previous versions. (This would also be a problem with querying a proprietary search engine like Google, but would not be a problem with an open source LLM or search engine.)

The second set of LLM experiments seem a bit more interesting. Here, starting from the set of genes discovered to be involved in sleep / social activity in their knockdown experiments, they prompt the model as to whether pairs of genes/proteins are interacting to modulate the behavior, asking the model to explain its behavior using the well-established "chain-of-thought" strategy. The result is that the model tries to reason, based on what is known about the input proteins, about how they might interact. The model hypothesizes at least one interaction not currently known in the literature, including between *mre11* and *DopEcR* and *Dop1R1*, which seems to have been verified in their experiments. However, I believe that the authors' claim that the model has "completely reveal[ed] knowns and unknowns for a scientific question," and that their work is the first attempt to do so, are vast overstatements. First, see Meta's Galactica model, on LLM for science (<https://arxiv.org/pdf/2211.09085.pdf>). Second, as described earlier, there is no guarantee that the model's outputs are grounded in reality; they must still be validated by a human. The authors did not seem to perform any validation on this set of outputs from the LLM, aside from the single interaction between *mre11* and *DopEcR/Dop1R1*. Due to my lack of expertise in this specific domain of biology, it is hard for me to validate the outputs myself, but one phenomenon I do immediately notice in the outputs (which is also a known behavior of chain-of-thought prompting) is the problem of confirmation bias: there are very few (potentially no?) examples where the model hypothesizes that the genes/proteins don't interact; in each case, it attempts to reason about why they would (likely at the cost of factual validity). Before making claims about anything being "completely revealed," I think it is important to perform a more analysis of errors, factuality, etc. of the model outputs. Assuming it is easier for a researcher to validate the reasoning provided by the LLM than it is for a researcher to come up with the reasoning itself, the approach does represent a promising strategy for accelerating science (the promise of this is well known, however, and the authors did not demonstrate e.g. improved efficiency empirically with a user study.) While this work is demonstrating an interesting application of LLMs for assisting in scientific discovery, the contributions in this work, at least on the side of LLM prompting and the usefulness of its outputs, are vastly over-claimed.^{[1][2][3]}

Smaller comments/questions:

- The authors describe the GPT-3 model in introduction but use a different model (ChatGPT), the details of which are proprietary and closed, misleading as to what the actual model used was; i.e. "to implement state-of-the-art GAIs, such as InstructGPT and ChatGPT, GPT-3 model was trained..." but GPT-3 is an older, different model from the ones used, and while the technologies are related the exact relationship is not clear.
- "Here we use the most powerful LLM developed thus far, ChatGPT" how are you defining powerful, here? is there a metric or qualitative evidence that you can use to make this claim more specific?
- LLM prediction accuracy: breakdown of true positive, false positive, true negative, and false negative predictions? Impact of known issues such as hallucination?
- How exactly was accuracy computed for the fly tracking experiments? Per-fly per-frame? What was the frame rate, and how many individual experiments (set of flies on a plate) correspond to the reported 21,600 manually evaluated frames? If calculated per-fly per-frame, I'm not sure that this is a great measure for actual impact of errors on the downstream analysis. A better evaluation of accuracy

would measure the difference between downstream measurements of interest — sleep, locomotor activity, etc — for the same fly/plate experiments obtained from the automatic method compared to a human (but I don't think this would be measured as accuracy).

- Typo on line 60: though -> thought

- line 278: "machine reasoning of LLMs were" -> "machine reasoning of LLMs was" or better, "LLMs were used..."

Detailed Responses to Reviewers' Comments

We thank the editor and the reviewers for their thoughtful comments and suggestions which have made the manuscript much stronger. We have addressed these comments and suggestions as described below. The original reviews are listed point-by-point. Our responses are in blue font. Edits made in the text of the manuscript are marked in red.

Reviewer #1:

1. This is an interesting set of studies that will be influential for shaping the direction of sleep research. The use of ChatGPT is, in some ways, a gimmick. However, the authors used a rationale strategy to identify genes and I found this approach to be refreshing. The sooner the field begins using these tools the better. I applaud them for taking this approach and believe that others will follow. I am also excited by evaluating sleep in individuals and in groups. There is a growing awareness that sleep is more plastic than previously thought. These types of studies have an opportunity to identify critical genes that may have been missed in previous screens. What these studies will tell us about sleep function is unclear as the approach is very new. Nonetheless, the approach will open the way for a better understanding of sleep function and regulation.

I do not have any major critiques. The paper is well written. The authors include all of the appropriate genetic controls. Moreover, the authors conduct a few studies to evaluate how a gene might alter pathways that could indirectly impact sleep (e.g. alternative interpretations (e.g. stress odors). Importantly, the authors interpretation of the data was measured and was consistent with the data presented.

Reply: Thank you for your comments. We are delighted to know that you find our work interesting and influential.

Reviewer #2:

1. Review Comment on “Large-language models completely reveal the molecular signatures regulating sleep and activity”

*Overall, the authors present an impressive study that utilizes a Large Language Model (LLM) to summarize relevant information and investigates the genetic basis of sleep, locomotor activity, and social behavior in flies. The development of a multiple fly video-tracing system and the large-scale behavior screen add valuable insights to the field. The finding that *mre11* regulates sleep specifically in a group condition is particularly noteworthy. However, there are some issues and areas that need further clarification.*

Reply: Thank you for your interest in our work and your constructive criticisms.

2. In the section on "Genome-wide LLM interpretation of the genetic basis of sleep, locomotor activity", for example, the authors mention that 12.5% of 42,794 fly protein isoforms were interpreted in sleep regulation (Fig. 1b and Supplementary Table 1). However, concerns arise regarding the accuracy and completeness of this interpretation, as only approximately 30% of the genes reported to be involved in sleep regulation, based on our incomplete collection, are found in the authors' list (Table S2A). To support the claims made in Fig 1c (also, line 86), it is important to provide references for the manually verified genes listed in Table S2A. Additionally, it is crucial to clarify the exact number of genes with positive responses and the accuracy of these responses. Since this is the first time the model has been used for predictions, more precise validations are necessary. Simply categorizing responses as "positive" without further detail is insufficient. Providing more comprehensive judgments would facilitate a better assessment of the method's effectiveness. Additionally, the accuracy of ChatGPT interpretation ranges from 78.3%-82.2% seems not accurate enough for some mature fields. The authors should mention the inadequacy of the ChatGPT, not only to exaggerate its advantages..

Reply: We agree with you and thus have performed additional analysis to carefully evaluate the performance of the genome-wide LLM interpretation. We collected 268, 283, and 49 fly genes reported to be involved in regulating sleep, locomotor and social activity, respectively, from the scientific literature. These genes were used as positive data to estimate Type II errors/false negative hits by directly calculating a standard measurement, sensitivity (S_n). For each behavior, we also randomly selected an equal number of genes from genes that have not been reported to be involved, and such a

resampling procedure was repeatedly performed for 20 times. Then, the average specificity (Sp) value was calculated to estimate Type I errors/false positive hits. The benchmark datasets including both positive and negative data are presented in Supplementary Data 2.

Based on these results, we found that the Sn values were calculated as 20.9%, 25.1%, and 18.4% for sleep, locomotor and social activity, respectively. This might be at least partially attributed to the fixed time frame for ChatGPT model training, as its knowledge was limited to data before 2022. However, the average Sp values were calculated as 92.8%, 92.9%, and 92.9% for sleep, locomotor and social activity, respectively, indicating a low false positive rate of ChatGPT interpretations. We agree that perhaps quite a few truly functional genes were missed, but in the cases when ChatGPT does provide a positive answer, it is of relatively high quality and thus still helpful for facilitating further experimental design. We have revised the relevant descriptions (marked in red) in the results section to mention both the inadequacy and the advantages of ChatGPT: “To critically evaluate the performance of ChatGPT for interpreting the 3 behaviors, we manually curated experimentally identified genes essential for sleep, locomotor and social activity from the scientific literature (Supplementary Data 2a-2c). In comparison with the prediction results of ChatGPT, a standard measurement, sensitivity (Sn), was calculated to estimate the false negative rate (Type II errors). To estimate the false positive rate (Type I errors), an equal number of genes were randomly selected from genes that have not been reported to regulate each of the behavior, and the average specificity (Sp) value was calculated after 20 rounds of resampling. For sleep, locomotor and social activity, the Sn values were 20.9%, 25.1%, and 18.4%, respectively (Fig. 1c; Supplementary Fig. 1a-1c), showing a high false negative rate of ChatGPT responses. This reflects the limitations of ChatGPT in searching information of this sort. Notably, the knowledge used for ChatGPT model training was limited to data before 2022 which also contributes partially to the high false negative rate. Indeed, an important sleep-regulating gene reported in 2022, *discs overgrown (dco/dbt)*¹⁹, was not recognized as a positive hit by ChatGPT (Supplementary Data 1a and 2a). For the 3 behaviors, the average Sp values ranged from 92.8%-92.9% (Fig. 1c; Supplementary Fig. 1a-1c), showing a low false positive rate in ChatGPT answers. Despite the high false negative rate, we believe this low false positive rate still supports the usefulness of LLMs in searching and summarizing

literature.” We have also included descriptions regarding these analyses in the methods section which are marked in red (“Benchmark data preparation” and “Performance evaluation”).

3. *In the section on “Development of multiple fly video-tracing instrument”, it would be beneficial to include more details about group fly sleep profiles, such as sleep curves, bout duration, bout number, and sleep latency in Extended Data Fig1a. Similarly, including sleep curves, bout duration, bout number, and sleep latency for the positive controls in Extended Data Fig2 would provide a better understanding of the results. Activity plot of pdfr mutants are required as its typical pattern in morning and evening anticipation. Furthermore, it is important to compare these profiles with previous studies to assess their consistency.*

Reply: Thank you for these wonderful suggestions. These analyses have now been added to Supplementary Fig. 3-7 and they are consistent with the published data for the most part.

4. *The supplementary videos showed error examples of video-to-location (VTL) strategy. Does the error percentage rise with the increasing number of flies?*

Reply: Yes, this a critical point. In this revised version, we performed additional analysis to monitor male and female flies with different group sizes ranging from 1 to 20. Then, we randomly selected 144 video fragments (7.2 frames/s) and manually counted the tracking errors in each of the 153,360 frames. The detailed data statistics for these procedures are shown in Supplementary Data 4. Then, the accuracy values were calculated for each group size. From these results, we found that the accuracy decreases significantly when the group size exceeds 5 individuals. Therefore for the remainder of the study, we only used data from groups with 5 or fewer individuals. We have added these results to Fig. 1g and Supplementary Fig. 2, and have removed the data for group sizes larger than 5.

5. *Regarding the comment about increased sleep and reduced locomotor activity often being caused by sickness, it is necessary to provide references supporting this claim*

(line 152-153). Given the different screening methods employed, particularly the group tracing approach, it is understandable that newly identified genes involved in sleep may differ from those reported in previous studies. However, it is surprising that only nine genes overlap between the 300+ genes identified in this study and the known genes (line 155-157). To address this, the authors should consider selecting more previously reported genes involved in sleep regulation as methodological positive controls or manipulating the neurons involved in sleep regulation to refine their tracking or experimental methods.

Reply: Thank you for these important points. We have now included a reference regarding increased sleep being caused by sickness in the text (PMID: 19209176). We have also tested more sleep-regulating genes (*Cul3*, *CanA-14F* and *Fmr1*) and activated octopaminergic neurons via two methods as additional positive controls (Supplementary Fig 6 and 7).

We apologize for the confusion caused by our previous statement regarding only 9 genes identified in our screen overlap with known sleep genes. These “known sleep genes” were defined by ChatGPT. We have now manually compiled and curated benchmark datasets based on published literature. There are 20 genes identified in our screen to regulate sleep that are also present in this benchmark dataset. We have modified the manuscript accordingly (Fig 2a).

6. In the section on “*mre11* and *NELF-B* regulate *Drosophila* sleep”, clarification is needed regarding the interaction frequency for *mre11* flies (Fig3i and j). Considering the possibility of a higher interaction number rather than duration disrupting fly sleep, this aspect should be addressed.

Reply: Thank you for this suggestion. We have now analyzed interaction number and found it to be reduced as well in *mre11* RNAi flies (Fig. 3j and 3l).

7. The authors propose that olfactory cues facilitate sleep when conspecifics are present and impaired sensory processing triggers sleep loss selectively under the group condition (line 210-212). However, this proposal appears inconsistent with the results

*presented in the study. Therefore, it is recommended to approach the interpretation of the results more cautiously based on the following points: (i) the significant reduction in sleep duration when five or more flies are grouped together suggests that olfactory cues do not promote sleep, at least in this context; (ii) previous studies have indicated that socially-enriched flies sleep more than socially-impooverished flies, suggesting that early social interaction might promote sleep (reference 4, activity tracing is isolated); (iii) for *mre11* RNAi flies, the mutant shows defects in olfactory response and group sleep only, which is inconsistent with the provided reference. If the reference is valid, *mre11* mutant flies should exhibit reduced sleep even under single tracing conditions, which is not observed. Therefore, a more reasonable interpretation could be that knockdown of *mre11* enhances some social interaction-related signals, strongly inhibiting sleep, rather than being a defect but a gain of function. This would explain why isolated tracing does not result in sleep reduction.*

Reply: Thank you for this wonderful comment. This issue has been puzzling us for quite a while and your explanation would make the most sense. We have now re-written this interpretation which reads as: “Taken together, these results implicate that lack of *mre11* leads to defective olfactory sensation. It is possible that when conspecifics are present, this altered olfactory sensation enhances some social-related signals that strongly inhibit sleep in *mre11* RNAi flies.”

8. *In Extended Data Fig.10, while the expression of some sleep-promoting genes is reduced, it is important to investigate whether the expression of sleep-inhibitory genes changed in these flies. Interpreting these results comprehensively would provide a more complete understanding of the molecular mechanisms underlying sleep regulation. Also, the authors should give more control and experiment group, such as UAS control or another RNAi line, to convince this result.*

Reply: Thank you for these suggestions. The genes that we have tested include both sleep-promoting and sleep-inhibiting genes. We have now organized the genes into these two categories (Supplementary Fig 15a and 15b). We have also added UAS controls (Supplementary Fig 15c).

9. In line 250-253 (and Extended fig15a), the observation that *Dop2R-GAL4>UNELF-B-RNAi* shows the largest reduction in sleep. It would be beneficial to include additional experimental data or analysis to validate function of *UNELF-B* in *Dop2R* neurons, as in Extended Data Fig 15b.

Reply: Thank you for this suggestion. We have knocked down *NELF-B* in *Dop2R* cells with the other two *NELF-B* RNAi lines but there is no significant effect on sleep. This has been added to Supplementary Fig 19b.

10. In the section on “Construction of a signal web via complete LLM-reasoning to demonstrate molecular signatures”.

Firstly, it appears that the 139 potential interactions depicted in Figure 4 do not overlap with the 17 protein interactions among documented in the BioGRID database. If these predictions are valid, it is crucial to explain why there is no correspondence between the predicted interactions and the protein interaction information in the BioGRID database. Clarifying this disparity would help establish the reliability and uniqueness of your findings.

Reply: We apologize for this confusion caused by our original definitions of gene-gene relationships reasoned by ChatGPT. We only queried ChatGPT regarding potential regulations between two genes, which includes indirect regulations and functional relations. We did not query ChatGPT regarding physical or genetic interactions between two genes because these information can be acquired from databases available. Moreover, some of these interactions in the databases are not described in words in the literature and thus this information cannot be obtained by ChatGPT. Therefore, here we used both ChatGPT and BioGRID together to generate the signal web. In this revised version, relationships without unambiguous regulatory directions were defined as “Functional associations”, while others with a clear direction of information flow were defined as “Regulations”. The type of regulation, positive or negative, was also indicated if available. For each relationship, we carefully searched PubMed, and found that 103 (74.1%) of the relationships were supported by literature. These primary references with PMIDs were added to Supplementary Data 8. The 17 protein or genetic PPIs obtained from the BioGRID database reflect relatively direct interaction, and thus

do not overlap with ChatGPT reasoning results. We have modified the results and methods sections accordingly (marked in red).

*11. Secondly, it would strengthen the effectiveness of their predictions if the authors could provide validation methods. For instance, validating interactions that influence gene expression, such as *mre11*'s regulation of *dop1R1* expression (as mentioned in Fig4a and Table S7C, despite inconsistency with Extended Data Fig10b), or employing alternative means to verify the accuracy of your predictions, would provide more confidence in the proposed molecular interactions. Including additional experimental evidence or validation techniques would be highly beneficial in bolstering the robustness of your conclusions.*

Reply: Thank you for this wonderful suggestion. We have now measured the mRNA levels of the 5 genes predicted to be targets of *mre11* or *NELF-B* and observed significant reduction for *Hdc* which is a predicted target of *mre11* (Supplementary Fig. 21). Furthermore, we treated *mre11* RNAi flies with drugs that target Dop1R and DopEcR (Fig. 4b-4i; Supplementary Fig. 22). We found that Dop1R antagonist SCH23390 rescues the reduced sleep bout number and partially rescues (effective on only one of the RNAi lines) the lengthened sleep latency in *mre11* RNAi flies (Fig. 4e and 4f).

*12. Lastly, the suggestion in lines 290-293, where LLM-reasoning is used to potentially support the regulation of dopamine receptors *DopEcR* and *Dop1R1* by *mre11*, which is consistent with previous pharmacological experiments, raises concerns. LLM-reasoning results cannot be further utilized to support the reliability of pharmacological results because the pharmacological results was used as the prompt information in the LLM-reasoning process.*

Reply: Thank you for pointing this out. We have removed this sentence.

Reviewer #3:

1. In this article the authors investigate the genetic regulation of sleep, locomotor and social activity in *Drosophila*. They do this by (1) using object tracking in videos to record the behavior of fruit flies as two-dimensional coordinates of individual flies in a culture plate while knocking down 5,588 genes, and (2) querying the ChatGPT large language model (LLM) whether fly protein isoforms are functionally involved in various behaviors (alone and in conjunction with each other). For the object tracking system, they used an out-of-the box object tracking model, manually evaluating 21,600 frames and reporting 98.58% accuracy. They performed 128,382 queries total, one for each of three behaviors (sleep, locomotor, and social?) * 42,794 fly protein isoforms to elicit predictions from the model as to the involvement of each protein in each behavior. They manually verified responses for known behaviors in the literature, estimating the model's accuracy at 78.3-82.2%. They also explore the *mre11* gene in more detail, which has not been previously reported to participate in sleep regulation.

Since my background/expertise is in machine learning and LLMs, my review will focus on those aspects of the article; I leave validation of the methodological soundness and novelty of the biology to other reviewers with more relevant expertise in that area.

Reply: We appreciate your precise descriptions of our work, and we now use “**object tracking in videos**” to describe our instrument in the revised manuscript (marked in red). We have also carefully revised the manuscript based on your comments by performing additional analyses as shown below.

2. The authors claim to propose a new prompting strategy for eliciting information from the LLM, which they call prime-boost prompt (PBP), to incorporate relevant background knowledge step-by-step into the prompts. The prompting strategy is not clearly described in the article, but the exact prompts used were shared in supplementary material, so I was able to look at them to try to discern more detail on their strategy. In both sets of prompting experiments, their strategy for eliciting information from the model is quite standard, applying known prompting templates to their specific domain of interest, and I do not think it warrants a special name (“PBP”). So, regarding novelty, the question is not whether the LLM prompting strategy is novel, but whether the use of the LLM as a tool enables novel analysis.

Reply: We agree with you and have removed the term “prime-boost prompt (PBP)” in the revised manuscript, which now reads as: “...To ensure the efficiency of prompt-generation response ¹⁸, we used **the standard** prompting strategy, which incorporates the background knowledge step by step into the full prompts to **elicit** ChatGPT to produce high-quality answers...”

3. The results of the first set of experiments prompting chatGPT are, in my opinion, not very interesting. In this set of experiments the authors ask the model whether a large set of fly proteins are involved in sleep and social activity. As one would expect, the model reflects a fairly accurate knowledge of what is known about these proteins from the scientific literature up until 2021 (the time cutoff for its training data.) The key takeaway here, which is well known, is that LLMs provide a human friendly interface for querying knowledge contained on the web. Pros of such an approach are that the output from a prompt (replacing a search query) is stated in natural english, essentially “Yes, ...” “No, ..” or “I don’t know..”, compared to search results from current search engines such as Google, which, for these types of queries, still respond with a list of relevant websites (in this case, mostly relevant research articles.) Flaws are that we can’t know for certain the veracity of the LM text, as current LLM technologies con’t accurately attribute their responses to sources/citations. So, this approach is useful for cases where such mistakes are ok. Knowing the relative rates of true positive / false positive / true negative / false negative predictions would be useful for making informed decisions about appropriate use cases for this approach.

Reply: We agree with you and thus have performed additional analysis to carefully evaluate the performance of the genome-wide LLM interpretation. We collected 268, 283, and 49 fly genes reported to be involved in regulating sleep, locomotor and social activity, respectively, from the scientific literature. These genes were used as positive data to estimate Type II errors/false negative hits by directly calculating a standard measurement, sensitivity (S_n). For each behavior, we also randomly selected an equal number of genes from genes that have not been reported to be involved, and such a resampling procedure was repeatedly performed for 20 times. Then, the average specificity (S_p) value was calculated to estimate Type I errors/false positive hits. The

benchmark datasets including both positive and negative data are presented in Supplementary Data 2.

Based on these results, we found that the S_n values were calculated as 20.9%, 25.1%, and 18.4% for sleep, locomotor and social activity, respectively. This might be at least partially attributed to the fixed time frame for ChatGPT model training, as its knowledge was limited to data before 2022. However, the average S_p values were calculated as 92.8%, 92.9%, and 92.9% for sleep, locomotor and social activity, respectively, indicating a low false positive rate of ChatGPT interpretations. We agree that perhaps quite a few truly functional genes were missed, but in the cases when ChatGPT does provide a positive answer, it is of relatively high quality and thus still helpful for facilitating further experimental design. We have revised the relevant descriptions (marked in red) in the results section which reads as: “To critically evaluate the performance of ChatGPT for interpreting the 3 behaviors, we manually curated experimentally identified genes essential for sleep, locomotor and social activity from the scientific literature (Supplementary Data 2a-2c). In comparison with the prediction results of ChatGPT, a standard measurement, sensitivity (S_n), was calculated to estimate the false negative rate (Type II errors). To estimate the false positive rate (Type I errors), an equal number of genes were randomly selected from genes that have not been reported to regulate each of the behavior, and the average specificity (S_p) value was calculated after 20 rounds of resampling. For sleep, locomotor and social activity, the S_n values were 20.9%, 25.1%, and 18.4%, respectively (Fig. 1c; Supplementary Fig. 1a-1c), showing a high false negative rate of ChatGPT responses. This reflects the limitations of ChatGPT in searching information of this sort. Notably, the knowledge used for ChatGPT model training was limited to data before 2022 which also contributes partially to the high false negative rate. Indeed, an important sleep-regulating gene reported in 2022, *discs overgrown (dco/dbt)*¹⁹, was not recognized as a positive hit by ChatGPT (Supplementary Data 1a and 2a). For the 3 behaviors, the average S_p values ranged from 92.8%-92.9% (Fig. 1c; Supplementary Fig. 1a-1c), showing a low false positive rate in ChatGPT answers. Despite the high false negative rate, we believe this low false positive rate still supports the usefulness of LLMs in searching and summarizing literature.” We have also included descriptions regarding these analyses in the methods section which are marked in red (“Benchmark data preparation” and “Performance evaluation”).

4. *Second, largely unlike current search engines, a limitation of current LLM technologies is that the responses are limited to a fixed time frame, based on when data was aggregated for LLM model training. In the case of ChatGPT, this is currently sometime during 2021, though eventually the model will be updated, and consequently get stale (without substantial advances in the area of continual/lifelong learning for LLMs.) On the other hand, new pages and articles can be quickly and easily indexed by current search systems nearly in real time.*

Reply: We agree with you and indeed we found that functional genes newly identified after 2021 were not interpreted by ChatGPT. For example, one sleep-regulating gene reported in 2022, *discs overgrown (dco/dbt)* was missed and we noted this in the results section (marked in red) which reads as: “Indeed, an important sleep-regulating gene reported in 2022, *discs overgrown (dco/dbt)*¹⁹, was not recognized as a positive hit by ChatGPT (Supplementary Data 1a and 2a).”

5. *Third, there are concerns about the replicability of the research, since the closed ChatGPT model was used, which is known to produce different results given the same prompt, due to the stochastic nature of model inference, and the underlying model changes over time, with no access to previous versions. (This would also be a problem with querying a proprietary search engine like Google, but would not be a problem with an open source LLM or search engine.)*

Reply: Yes, we have also observed that different results will be provided for the same prompt. However, the conclusions from these different responses are highly similar. To ensure the validity of this study, we have included all prompts and responses in Supplementary Data 1.

6. *The second set of LLM experiments seem a bit more interesting. Here, starting from the set of genes discovered to be involved in sleep / social activity in their knockdown experiments, they prompt the model as to whether pairs of genes/proteins are interacting to modulate the behavior, asking the model to explain its behavior using the*

well-established “chain-of-thought” strategy. The result is that the model tries to reason, based on what is known about the input proteins, about how they might interact. The model hypothesizes at least one interaction not currently known in the literature, including between mre11 and DopEcR and Dop1R1, which seems to have been verified in their experiments. However, I believe that the authors’ claim that the model has “completely reveal[ed] knowns and unknowns for a scientific question,” and that their work is the first attempt to do so, are vast overstatements. First, see Meta’s Galactica model, on LLM for science (<https://arxiv.org/pdf/2211.09085.pdf>).

Reply: We agree with you and the word “completely” has been removed when describing ChatGPT interpretation or reasoning. We have carefully read the Galactica paper which implemented an attractive LLM model for science. Yet, how to use LLMs to facilitate scientific discovery has not yet been demonstrated, and we believe our study is the first attempt for this purpose. We have modified the relevant discussion which now reads as: “To the best of our knowledge, our study is the first attempt that **demonstrates the application of LLMs for assisting scientific discovery.**” We have also changed the title of our paper to “Large-language models **facilitate discovery of** the molecular signatures regulating sleep and activity”.

7. Second, as described earlier, there is no guarantee that the model’s outputs are grounded in reality; they must still be validated by a human. The authors did not seem to perform any validation on this set of outputs from the LLM, aside from the single interaction between mre11 and DopEcR/Dop1R1. Due to my lack of expertise in this specific domain of biology, it is hard for me to validate the outputs myself, but one phenomenon I do immediately notice in the outputs (which is also a known behavior of chain-of-thought prompting) is the problem of confirmation bias: there are very few (potentially no?) examples where the model hypothesizes that the genes/proteins don’t interact; in each case, it attempts to reason about why they would (likely at the cost of factual validity). Before making claims about anything being “completely revealed,” I think it is important to perform a more analysis of errors, factuality, etc. of the model outputs.

Reply: Thank you for these comments. For each relationship reasoned by ChatGPT, we carefully searched PubMed and found that 103 (74.1%) of the relationships were

supported by literature. These primary references with PMIDs were added to Supplementary Data 8. In addition, we have experimentally validated the outputs for *mre11* and *NELF-B*. We measured the mRNA levels of the 5 genes predicted to be targets of *mre11* or *NELF-B* and observed significant reduction for *Hdc* which is a predicted target of *mre11* (Supplementary Fig. 21). Furthermore, we treated *mre11* RNAi flies with drugs that target Dop1R and DopEcR (Fig. 4b-i; Supplementary Fig. 22). We found that Dop1R antagonist SCH23390 rescues the reduced sleep bout number and partially rescues (effective on only one of the RNAi lines) the lengthened sleep latency in *mre11* RNAi flies (Fig. 4e and 4f). These results imply *Hdc* and Dop1R as down-stream signaling targets of MRE11 to modulate sleep, locomotor and social activity.

Regarding why there is no prediction regarding genes that do not interact, this is because in the great majority of the cases two genes do not interact with each other. We assume all genes that have not been predicted to interact do not interact. Of course, for some of the cases, two genes actually will be proved to interact in future studies under circumstances that have not been tested currently. This is why in biology it is relatively easy to demonstrate the presence of interaction but almost impossible to demonstrate that there is no interaction (for it is difficult to draw conclusion from negative data because as technology advances or as our knowledge advances, these negative data may no longer be negative). Of course we realize since we do not provide information regarding genes that do not interact, we certainly cannot use the word “complete”. We have removed it when describing ChatGPT interpretation or reasoning.

8. Assuming it is easier for a researcher to validate the reasoning provided by the LLM than it is for a researcher to come up with the reasoning itself, the approach does represent a promising strategy for accelerating science (the promise of this is well known, however, and the authors did not demonstrate e.g. improved efficiency empirically with a user study.) While this work is demonstrating an interesting application of LLMs for assisting in scientific discovery, the contributions in this work, at least on the side of LLM prompting and the usefulness of its outputs, are vastly over-claimed.^[SEP]

Reply: Thank you very much for your suggestions. We have now included descriptions regarding the improved efficiency of ChatGPT vs. manual curation in the discussion with reads as: “In this study, we found that answering 3,655 questions by ChatGPT only took about 3 hr, with ~3 s/answer. On the other hand, we manually curated the experimental evidence of 139 pairs of potential functional regulations or associations from the literature for over one week (~10 hr/day), with ~30 min/answer. Thus, ChatGPT can dramatically promote our research efficiency.” We agree with you regarding the over-claim and have modified the manuscript accordingly as described in previous responses.

9. *Smaller comments/questions:*

The authors describe the GPT-3 model in introduction but use a different model (ChatGPT), the details of which are proprietary and closed, misleading as to what the actual model used was; i.e. “to implement state-of-the-art GAIs, such as InstructGPT and ChatGPT, GPT-3 model was trained...” but GPT-3 is an older, different model from the ones used, and while the technologies are related the exact relationship is not clear.

Reply: Thank you for your comments. We have modified this description (marked in red) which now reads as: “One of the state-of-the-art GAIs, ChatGPT/GPT 3.5, was trained with 175 billion parameters, and then fine-tuned using reinforcement learning from human feedback to align language models with human chains of thought (CoTs) ^{13,14,15}.”

10. *“Here we use the most powerful LLM developed thus far, ChatGPT” how are you defining powerful, here? is there a metric or qualitative evidence that you can use to make this claim more specific?*

Reply: We apologize for the inappropriate description and have modified the text which now reads as: “Here we used the release of ChatGPT/GPT 3.5 to help us address the regulatory mechanism of sleep, locomotor and social activities.”

11. *LLM prediction accuracy: breakdown of true positive, false positive, true negative, and false negative predictions? Impact of known issues such as hallucination?*

Reply: Thank you for these comments. Please see our response to your third point regarding these issues.

12. *How exactly was accuracy computed for the fly tracking experiments? Per-fly per-frame? What was the frame rate, and how many individual experiments (set of flies on a plate) correspond to the reported 21,600 manually evaluated frames? If calculated per-fly per-frame, I'm not sure that this is a great measure for actual impact of errors on the downstream analysis. A better evaluation of accuracy would measure the difference between downstream measurements of interest — sleep, locomotor activity, etc — for the same fly/plate experiments obtained from the automatic method compared to a human (but I don't think this would be measured as accuracy).*

Reply: Yes, we calculated the accuracy for the fly tracking by counting per-fly per-frame. The frame rate is 7.2 frames/s in our study. In this revised version, we performed an additional analysis to monitor male and female flies with different group sizes ranging from 1 to 20. Then, we randomly selected 144 video fragments (7.2 frames/s), and manually counted the tracking errors in each of the 153,360 frames. It took two authors, Dan Liu and Miaoying Zhao, about 2 weeks to conduct the first round of calculation, with about 10,000 frames analyzed per day. Then two additional students, Liubin Zheng and Cheng Han, conducted a double-check which took another week. The detailed data statistics for these procedures are shown in Supplementary Data 4. Then, the accuracy values were calculated for each group size. From the results, we found that the accuracy decreases significantly when the group size exceeds 5 individuals. Therefore for the remainder of the study, we only used data from groups with 5 or fewer individuals. We have added these results to Fig. 1g and Supplementary Fig. 2, and have removed the data for group sizes larger than 5.

8. *Typo on line 60: though -> thought*

Reply: We have corrected this typo.

9. line 278: “*machine reasoning of LLMs were*” -> “*machine reasoning of LLMs was*”
or better, “*LLMs were used...*”

Reply: We have rephrased this description to “...LLMs were used...”

REVIEWER COMMENTS

Reviewer #1 (Remarks to the Author):

I was enthusiastic about this work and remain so. I am satisfied with the authors responses and I have no additional concerns.

Reviewer #2 (Remarks to the Author):

The manuscript has been significantly improved with a few minor points to be addressed:

1. In Response to Referee's Letter (Reviewer #2), "Point 2":

For Supplementary Figures 1a-1c, please change the vertical axis label from "actual" to "reported."

2. In Response to Referee's Letter (Reviewer #2), "Point 3":

Regarding Supplementary Figure 3, it is interesting to note that *Drosophila*'s sleep duration decreased with an increase in the number of flies, while the sleep bout number increased and sleep bout duration decreased. This suggests that *Drosophila* exhibit more fragmented sleep in a group condition. If possible, consider adding some comments on this phenomenon in the "Discussion" section (e.g., how odors or sounds around flies may affect their sleep) to better highlight your findings. This is an optional suggestion.

For Supplementary Figure 5 and other figures, please label the "control" group with the exact genotype, such as "WT," "iso CS," or "W1118," as we do not have the exact information on what "control" means, and it could be misleading.

Regarding lines 165-168, are these sleep profile details, such as bout duration/number, consistent with previous reports? Please describe these results as you did in lines 168-171 for the phenotype of Pdf^{rhan5304}.

3. In Response to Referee's Letter (Reviewer #2), "Point 5":

The reference (PMID:19209176) provided by the author to support the statement that "increased sleep and reduced locomotor activity are often caused by sickness" is inappropriate. In this review paper (PMID:19209176), the discussion centers on how and why infection (or sickness) alters sleep, rather than providing evidence to support that increased sleep indicates sickness in mice. Moreover, in Figure 4, they summarize that infection decreases REM sleep rather than promoting it. In other examples, the loss of function of genes such as *pdf* and *pdfr* increases sleep, and gain of function mutation in *SIK3* (sleepy) also increases sleep. To our knowledge, there is no clear evidence to support the claim that these flies are sick. Therefore, we strongly recommend removing this comment or changing "often" to "sometimes" if you believe it is crucial for your paper to make this claim. The statement "Increased sleep and/or reduced locomotor activity are often caused by sickness" can be misleading.

It is interesting to note that there is a low overlap rate, as only 20 genes identified in your screen to regulate sleep are also present in this benchmark dataset. If possible, consider adding some comments on this phenomenon in the "Discussion" section.

4. In Response to Referee's Letter (Reviewer #2), "Point 7":

Regarding the rewritten hypothesis on how MRE11 may impact sleep as "this altered olfactory sensation enhances some social-related signals that strongly inhibit sleep in MRE11 RNAi flies," I have another concise hypothesis to explain your results: "MRE11 is essential for flies to recognize conspecifics, and the loss of MRE11 function impairs this process, such as the disruption of recognizing pheromones from conspecifics or other information through olfactory sensation. This disruption causes

flies to mistake conspecifics for heterospecifics, inducing a 'nervous' or stressed state that disrupts sleep in group condition. In isolated condition, there is no such stress, so they sleep as much as wild-type flies." Importantly, this hypothesis can be easily tested by grouping *Drosophila melanogaster* with other *Drosophila* species under MRE11 knockdown or wild-type conditions, which could provide preliminary evidence. I strongly recommend, although this is an optional suggestion for this paper, that you test this hypothesis. If the results are positive, it will highlight at least two points: 1) Your multiple flies tracing system for researching the impact of heterospecifics on sleep or similar research situations. 2) The mechanism of MRE11 on sleep. This could potentially open up exciting possibilities for further investigation into the function of MRE11.

5. In Response to Referee's Letter (Reviewer #2), "Point 8":

In lines 282-283, since there is no evidence supporting that MRE11 acts upon Hk or Ilp3, we recommend revising the sentence "These are potential candidates that MRE11 acts upon to promote sleep" to "There is a possibility that MRE11 may somehow affect Hk or Ilp3 to promote sleep."

6. In response to Referee's Letter (Reviewer #2), "Point 11":

There is a design flaw in this experiment. Specifically, in lines 347-352, Figure 4e-4i, and related supplementary figures, only the differences between different genotype groups are considered under the condition that all genotype flies are fed with inhibitors. However, it fails to address whether Dop1R is the primary mediator through which MRE11 regulates sleep. In other words, it does not investigate whether the administration (i.e., blocking Dop1R) affects the sleep impact caused by MRE11 knockdown specifically in the context of MRE11 knockdown.

Regarding lines 352-354, the authors have provided evidence thus far to support that MRE11 affects *Drosophila* sleep, and MRE11 knockdown alters the expression of Dop1R and Hdc, which are known to be involved in sleep regulation. However, there is no evidence to establish the upstream-downstream relationship between MRE11 and Dop1R or Hdc. It would be more appropriate to summarize these results as follows: "These findings suggest that MRE11 may influence *Drosophila* sleep by modulating the expression of Dop1R and Hdc in an unknown manner."

Reviewer #3 (Remarks to the Author):

Overall, the authors provided satisfactory edits, additional details and experiments in response to my concerns; I appreciate their willingness to make many edits following my and the other reviewers' suggestions. I provide further comments/feedback on this draft (mostly related to writing clarity) below.

I think it should be made clear very early in the paper (abstract/intro) the connection between the LLM and video tracking / gene expression experiments. In the current draft no connection is made until line ~309. Until then the article reads to me as if they are completely distinct sets of experiments which is somewhat confusing (and would require a different title, maybe more generally referring to AI methods rather than LLMs specifically). The article would benefit not only from moving this information earlier in the article, but also motivating the connection/need for both methods more clearly. A good place to clarify this would be by editing lines 63-65 to be more specific about how GPT3.5 is being used in relation to the regulatory mechanism ("help us address" is too vague a phrase that provides no concrete indication of how the model will be used.) This is also where mention of the video tracking experiments should come in (in relation to the LLM); it is jarring that the video tracking experiments aren't mentioned in the introduction, then many pages of results are only talking about those experiments. Similarly, the abstract doesn't clearly connect the two sets of experiments.

As the article is currently written, it seems as though the authors do not really understand how the LLM model works. I am guessing that none of the authors is very familiar with the underlying technology (i.e. an NLP expert). This is ok, but here is some feedback on how the discussion of the LLM could be improved for accuracy and clarity:

- Line 57: In the context of this article, which deals primarily with LLMs ability to effectively encode information relevant to drosophila sleep, locomotor and social behavior, it's not clear why the number of model parameters is included. Instead, any available information about the training data would be more relevant here. Later, it is mentioned that the model was trained on data up until 2022, yet in this earlier introduction to the model no mention of the training data is made. I am aware that unfortunately OpenAI is not forthcoming with this information, but I think anything that is known about the quantity and type of text and instructions used for pretraining and finetuning would be relevant to include here, and more relevant than the number of model parameters.
- Line 59: "Due to the complexity of LLMs," part is not really accurate, I would rewrite this sentence to something closer to "Prompt engineering has been demonstrated as an effective strategy for eliciting information from LLMs"
- Lines 57-64 describing the specific model used are still unclear. If you used GPT3.5, please just say GPT3.5, or whatever specific model was used.
- Obscure nonstandard citations are included in a number of places while important citations are missing. For example, Knox and Stone (2011), while having a relevant title, is a bit of a strange and nonstandard citation, particularly since they do not experiment with text in the article. You might instead include missing citations on chain-of-thought prompting (this really should be included given that you even use the "chain-of-thought" terminology introduced in that paper), e.g. Wei et al. 2022, "Chain of Thought Prompting Elicits Reasoning in Large Language Models"; and RLHF, e.g. Christiano et al (2017), "Deep reinforcement learning from human preferences". Also, Strobel et al. 2022 is a strange citation to include when introducing the prompting strategy; I recommend including a more standard citation such as the chain-of-thought paper or the original GPT-3 "Language models are few-shot learners" paper, one of the earlier papers where prompting was introduced and successfully used.
- line 72: the -> a (while the strategy is standard, it is one among many)
- line 381: be sure to refer to the model the same way here as in intro (i.e. if you say GPT3.5 in intro, say it here as well.)

It would be fantastic if you could include an example prompt and response in the article. I realize they are quite long, and available in supplementary material, but it would be very helpful for understanding the methodology (could devise a abbreviated example). I realize this may be outside the scope of the formatting of the journal but I highly recommend an overview figure is included with a demonstrative example.

Typos:

- line 35: question -> questions

Detailed Responses to Reviewers' Comments

We thank the reviewers for their thoughtful comments and suggestions which have made the manuscript much stronger. We have addressed these comments and suggestions as described below. The original reviews are listed point-by-point. Our responses are in blue font. Edits made in the text of the manuscript are marked in red.

Reviewer #1:

I was enthusiastic about this work and remain so. I am satisfied with the authors responses and I have no additional concerns.

Reply: Thank you for your appreciation.

Reviewer #2:

The manuscript has been significantly improved with a few minor points to be addressed:

1. In Response to Referee's Letter (Reviewer #2), "Point 2":

For Supplementary Figures 1a-1c, please change the vertical axis label from "actual" to "reported."

Reply: This has been modified.

2. In Response to Referee's Letter (Reviewer #2), "Point 3":

*Regarding Supplementary Figure 3, it is interesting to note that *Drosophila*'s sleep duration decreased with an increase in the number of flies, while the sleep bout number increased and sleep bout duration decreased. This suggests that *Drosophila* exhibit more fragmented sleep in a group condition. If possible, consider adding some comments on this phenomenon in the "Discussion" section (e.g., how odors or sounds around flies may affect their sleep) to better highlight your findings. This is an optional suggestion.*

Reply: Thank you for bringing up this point. We have added a paragraph at the beginning of the Discussion (marked in red) which reads as: “Liu et al. reported substantially reduced daily sleep duration and sleep bout length accompanied by increased sleep bout numbers for fly populations containing 50 individuals compared to fly recorded in isolation⁶. Here, we observe sleep fragmentation with as few as 2 flies in a group, while daily sleep duration displays significant shortening when group size reaches 3 individuals. This demonstrates the prominent influence of social signal-related sensory cues on sleep quantity and quality.”

3. *For Supplementary Figure 5 and other figures, please label the "control" group with the exact genotype, such as "WT," "iso CS," or "W1118," as we do not have the exact information on what "control" means, and it could be misleading.*

Reply: Thank you for pointing this out. This has been edited.

4. *Regarding lines 165-168, are these sleep profile details, such as bout duration/number, consistent with previous reports? Please describe these results as you did in lines 168-171 for the phenotype of Pdf¹RNAi.*

Reply: Thank you for this suggestion. We have now included descriptions comparing these sleep parameters with previous publications where applicable (marked in red), which reads as: “We further compared sleep parameters of these flies in our group condition with the published work. We found a trend of reduction in sleep bout length and lengthened sleep latency in *Cul3* RNAi flies, as well as shortened sleep bout length in *CanA-14F* RNAi flies, in agreement with what has been reported in the literature^{30, 32}. We observed decreased sleep bout duration and sleep bout number in *Fmr1* over-expressing flies, similar with the previous study³¹.” Detailed sleep parameters were not reported for activation of octopaminergic neurons.

5. *In Response to Referee's Letter (Reviewer #2), "Point 5":*

The reference (PMID:19209176) provided by the author to support the statement that "increased sleep and reduced locomotor activity are often caused by sickness" is inappropriate. In this review paper (PMID:19209176), the discussion centers on how and why infection (or sickness) alters sleep, rather than providing evidence to support that increased sleep indicates sickness in mice. Moreover, in Figure 4, they summarize that infection decreases REM sleep rather than promoting it. In other examples, the loss of function of genes such as pdf and pdfR increases sleep, and gain of function mutation in SIK3 (sleepy) also increases sleep. To our knowledge, there is no clear evidence to support the claim that these flies are sick. Therefore, we strongly recommend removing this comment or changing "often" to "sometimes" if you believe it is crucial for your paper to make this claim. The statement "Increased sleep and/or reduced locomotor activity are often caused by sickness" can be misleading.

Reply: We have removed this sentence.

6. It is interesting to note that there is a low overlap rate, as only 20 genes identified in your screen to regulate sleep are also present in this benchmark dataset. If possible, consider adding some comments on this phenomenon in the "Discussion" section.

Reply: We have discussed about the possible reasons in the second paragraph of Discussion, including insufficient RNAi knock-down efficiency and novel sleep-regulating genes specifically associated with the social environment. We have now included an additional possibility (marked in red): “On the other hand, some of the genes that regulate sleep under isolated condition may be dispensable under group condition.”

7. In Response to Referee's Letter (Reviewer #2), "Point 7":

Regarding the rewritten hypothesis on how MRE11 may impact sleep as "this altered olfactory sensation enhances some social-related signals that strongly inhibit sleep in MRE11 RNAi flies," I have another concise hypothesis to explain your results: "MRE11 is essential for flies to recognize conspecifics, and the loss of MRE11 function impairs this process, such as the disruption of recognizing pheromones from conspecifics or

other information through olfactory sensation. This disruption causes flies to mistake conspecifics for heterospecifics, inducing a 'nervous' or stressed state that disrupts sleep in group condition. In isolated condition, there is no such stress, so they sleep as much as wild-type flies." Importantly, this hypothesis can be easily tested by grouping *Drosophila melanogaster* with other *Drosophila* species under *MRE11* knockdown or wild-type conditions, which could provide preliminary evidence. I strongly recommend, although this is an optional suggestion for this paper, that you test this hypothesis. If the results are positive, it will highlight at least two points: 1) Your multiple flies tracing system for researching the impact of heterospecifics on sleep or similar research situations. 2) The mechanism of *MRE11* on sleep. This could potentially open up exciting possibilities for further investigation into the function of *MRE11*.

Reply: Thank you for this brilliant suggestion. We have now monitored each *mre11* RNAi fly and controls with another conspecific (*Drosophila melanogaster*) or a heterospecific (*Drosophila simulans*). However, daily sleep duration is substantially shortened regardless of the presence of a conspecific or a heterospecific, along with decreased sleep bout duration (Supplementary Fig. 12). These results suggest that the sleep inhibition caused by *mre11* deficiency is not due to mis-recognition of conspecifics as heterospecifics.

8. In Response to Referee's Letter (Reviewer #2), "Point 8":

In lines 282-283, since there is no evidence supporting that *MRE11* acts upon *Hk* or *Ilp3*, we recommend revising the sentence "These are potential candidates that *MRE11* acts upon to promote sleep" to "There is a possibility that *MRE11* may somehow affect *Hk* or *Ilp3* to promote sleep."

Reply: Thank you for bringing up this point. We have now revised this sentence to: "These results indicate a possibility that *MRE11* may somehow affect *Hk* and/or *Ilp3* to promote sleep."

9. In response to Referee's Letter (Reviewer #2), "Point 11":

There is a design flaw in this experiment. Specifically, in lines 347-352, Figure 4e-4i, and related supplementary figures, only the differences between different genotype groups are considered under the condition that all genotype flies are fed with inhibitors. However, it fails to address whether Dop1R is the primary mediator through which MRE11 regulates sleep. In other words, it does not investigate whether the administration (i.e., blocking Dop1R) affects the sleep impact caused by MRE11 knockdown specifically in the context of MRE11 knockdown.

Reply: Thank you for this critical comment. We have now compared the effects of SCH23390 treatment on each genotype, and found that SCH23390 significantly shortens sleep latency in *mre11* RNAi flies but not the controls (Supplementary Fig. 24).

10. Regarding lines 352-354, the authors have provided evidence thus far to support that MRE11 affects Drosophila sleep, and MRE11 knockdown alters the expression of Dop1R and Hdc, which are known to be involved in sleep regulation. However, there is no evidence to establish the upstream-downstream relationship between MRE11 and Dop1R or Hdc. It would be more appropriate to summarize these results as follows: "These findings suggest that MRE11 may influence Drosophila sleep by modulating the expression of Dop1R and Hdc in an unknown manner."

Reply: Thank you for bringing up this point. We have now revised this sentence to: **“These findings suggest that MRE11 may influence sleep, locomotor and social activity by modulating *Dop1R* and *Hdc* in an unknown manner.”**

Reviewer #3:

1. Overall, the authors provided satisfactory edits, additional details and experiments in response to my concerns; I appreciate their willingness to make many edits following my and the other reviewers' suggestions. I provide further comments/feedback on this draft (mostly related to writing clarity) below.

I think it should be made clear very early in the paper (abstract/intro) the connection between the LLM and video tracking / gene expression experiments. In the current draft

no connection is made until line ~309. Until then the article reads to me as if they are completely distinct sets of experiments which is somewhat confusing (and would require a different title, maybe more generally referring to AI methods rather than LLMs specifically). The article would benefit not only from moving this information earlier in the article, but also motivating the connection/need for both methods more clearly. A good place to clarify this would be by editing lines 63-65 to be more specific about how GPT3.5 is being used in relation to the regulatory mechanism (‘‘help us address’’ is too vague a phrase that provides no concrete indication of how the model will be used.) This is also where mention of the video tracking experiments should come in (in relation to the LLM); it is jarring that the video tracking experiments aren’t mentioned in the introduction, then many pages of results are only talking about those experiments. Similarly, the abstract doesn’t clearly connect the two sets of experiments.

Reply: We fully agree with you, and revised the abstract to strengthen the connection between LLM interpretation and video tracking. We have also added another paragraph to the Introduction, which reads as: ‘‘Here, we first conduct a genome-wide interpretation of the genetic basis of sleep, locomotor and social activity regulation in *Drosophila melanogaster*, using a standard prompting strategy to elicit information from GPT-3.5. 12.5%, 13.8%, and 10.2% of the fly protein isoforms are interpreted to be involved in sleep, locomotor and social activity regulation, respectively, with low false positive rates. These numbers demonstrate the usefulness of GPT-3.5 in collection and summarization of relevant information. In parallel, we develop a video-tracking instrument to simultaneously monitor the real-time behavior of multiple fruit flies. Using this system, we conduct a genome-wide RNA interference (RNAi) screen, and identify 285, 310, and 359 genes to be potentially involved in regulating sleep, locomotor and social activity, respectively. Besides a number of genes recognized by GPT-3.5 to regulate these three behaviors based on published literature, we also identify many more that have not been previously reported, such as *mre11* and *NELF-B*. Then, an educated signaling network is modeled for 86 candidates from the screen, using the CoT prompting strategy of LLM-reasoning to pairwise identify potential functional regulations or associations among these genes. We further validate the potential mechanisms reasoned by LLM and reveal that MRE11 may influence sleep, locomotor and social activity by modulating dopamine receptor *Dop1R1* and *Histidine*

*decarboxylase (Hdc). In summary, here we systematically analyze the molecular mechanism regulating sleep, locomotor and social activities by utilizing *in silico* interpretation and reasoning from LLMs-generated contextual information, in combination with genetic screens using our multi-object video-tracking paradigm. We anticipate that such a human-LLM interactive practice can be readily adopted to investigate other complex scientific questions as well.”*

2. As the article is currently written, it seems as though the authors do not really understand how the LLM model works. I am guessing that none of the authors is very familiar with the underlying technology (i.e. an NLP expert). This is ok, but here is some feedback on how the discussion of the LLM could be improved for accuracy and clarity:

- Line 57: In the context of this article, which deals primarily with LLMs ability to effectively encode information relevant to drosophila sleep, locomotor and social behavior, itâs not clear why the number of model parameters is included. Instead, any available information about the training data would be more relevant here. Later, it is mentioned that the model was trained on data up until 2022, yet in this earlier introduction to the model no mention of the training data is made. I am aware that unfortunately OpenAI is not forthcoming with this information, but I think anything that is known about the quantity and type of text and instructions used for pretraining and finetuning would be relevant to include here, and more relevant than the number of model parameters.

- Line 59: âDue to the complexity of LLMs,â part is not really accurate, I would rewrite this sentence to something closer to âPrompt engineering has been demonstrated as an effective strategy for eliciting information from LLMsâ

- Lines 57-64 describing the specific model used are still unclear. If you used GPT3.5, please just say GPT3.5, or whatever specific model was used.

- Obscure nonstandard citations are included in a number of places while important citations are missing. For example, Knox and Stone (2011), while having a relevant title, is a bit of a strange and nonstandard citation, particularly since they do not experiment with text in the article. You might instead include missing citations on

chain-of-thought prompting (this really should be included given that you even use the “chain-of-thought” terminology introduced in that paper), e.g. Wei et al. 2022, “Chain of Thought Prompting Elicits Reasoning in Large Language Models” ; and RLHF, e.g. Christiano et al (2017), “Deep reinforcement learning from human preferences” . Also, Strobelt et al. 2022 is a strange citation to include when introducing the prompting strategy; I recommend including a more standard citation such as the chain-of-thought paper or the original GPT-3 “Language models are few-shot learners” paper, one of the earlier papers where prompting was introduced and successfully used.

Reply: Thank you for your suggestions. The corresponding text has been carefully revised according to your comments, and now reads as: “...In order to develop GPT-3, the architecture of Transformer neural network using an attention or self-attention mechanism was adopted¹³. For model training, GPT-3 used a combination of five data sets, including a filtered monthly CommonCrawl data covering 2016 to 2019, an expanded release of the WebText dataset (WebText2), two internet-based books corpora (Books1 and Books2) and English-language Wikipedia, approximately equivalent to 500 billion byte-pair-encoded tokens¹⁴. Later, GPT-3 was fine-tuned into the InstructGPT model by conducting reinforcement learning from human feedback (RLHF)^{15,16}. Besides the InstructGPT dataset, a dialogue dataset, not clearly reported by OpenAI, was generated from conversations between human AI trainers and the chatbot, which was further fine-tuned to the state-of-the-art model, ChatGPT or formally GPT 3.5 (<https://openai.com/blog/chatgpt>). To solve complex tasks, prompt engineering has been demonstrated as an effective strategy for eliciting information from LLMs^{14,17}. In particular, complex reasoning in LLMs can be elicited by using a method termed chain-of-thought (CoT) prompting¹⁷....”

3. - line 72: the -> a (while the strategy is standard, it is one among many)

Reply: This has been edited.

4. - line 381: be sure to refer to the model the same way here as in intro (i.e. if you say GPT3.5 in intro, say it here as well.)

Reply: This has been fixed.

5. It would be fantastic if you could include an example prompt and response in the article. I realize they are quite long, and available in supplementary material, but it would be very helpful for understanding the methodology (could devise a abbreviated example). I realize this may be outside the scope of the formatting of the journal but I highly recommend an overview figure is included with a demonstrative example.

Reply: Thank you for this suggestion. We have now included an example of a standard prompting and a CoT prompting in Fig.5, along with descriptions in Discussion (3rd paragraph, marked in red).

6. Typos:

- line 35: question -> questions

Reply: This has been fixed.

REVIEWERS' COMMENTS

Reviewer #2 (Remarks to the Author):

Reviewer #2:

The manuscript has undergone significant improvements, and I am satisfied with the authors' response as most of my concerns have been addressed. There is only one minor point that I recommend modifying (see below). I believe there is no need for another round of review.

7. In Response to Referee's Letter (Reviewer #2), "Point 7":

Regarding the rewritten hypothesis on how MRE11 may impact sleep as "this altered olfactory sensation enhances some social-related signals that strongly inhibit sleep in MRE11 RNAi flies," I have another concise hypothesis to explain your results: "MRE11 is essential for flies to recognize conspecifics, and the loss of MRE11 function impairs this process, such as the disruption of recognizing pheromones from conspecifics or other information through olfactory sensation. This disruption causes flies to mistake conspecifics for heterospecifics, inducing a 'nervous' or stressed state that disrupts sleep in group condition. In isolated condition, there is no such stress, so they sleep as much as wild-type flies." Importantly, this hypothesis can be easily tested by grouping *Drosophila melanogaster* with other *Drosophila* species under MRE11 knockdown or wild-type conditions, which could provide preliminary evidence. I strongly recommend, although this is an optional suggestion for this paper, that you test this hypothesis. If the results are positive, it will highlight at least two points: 1) Your multiple flies tracing system for researching the impact of heterospecifics on sleep or similar research situations. 2) The mechanism of MRE11 on sleep. This could potentially open up exciting possibilities for further investigation into the function of MRE11.

Reply: Thank you for this brilliant suggestion. We have now monitored each *mre11* RNAi fly and controls with another conspecific (*Drosophila melanogaster*) or a heterospecific (*Drosophila simulans*). However, daily sleep duration is substantially shortened regardless of the presence of a conspecific or a heterospecific, along with decreased sleep bout duration (Supplementary Fig. 12). These results suggest that the sleep inhibition caused by *mre11* deficiency is not due to mis-recognition of conspecifics as heterospecifics.

The interpretation needs to be revised, marked in purple.

I understand that the results of Fig S12 did not support the notion that sleep inhibition caused by *mre11* deficiency is not due to misrecognition of conspecifics as heterospecifics. It appears that the interpretation is incorrect. In Fig S12 o-r, except for sleep latency, the sleep parameters of the three control groups of flies vary in the presence of D.m or D.s, making it challenging to claim that sleep parameters are affected by the presence of heterospecifics. This is a prerequisite for discussing whether *mre11* deficiency affects sleep due to misrecognition of conspecifics as heterospecifics or not. Considering that the sleep duration of *mre11*-deficient flies is even less in the presence of D.s compared to the presence of D.m (Fig S12 o,p), the authors' claim that "the sleep inhibition caused by *mre11* deficiency is not due to misrecognition of conspecifics as heterospecifics" should be correct. (If sleep inhibition caused by *mre11* deficiency is due to misrecognition of conspecifics as heterospecifics, the sleep duration of these flies in the presence of D.m or D.s should be comparable.)

The social interaction parameters of all control flies in Fig S12 t-u are consistent, indicating that the presence of heterospecifics suppresses the fly's social interaction control to conspecifics. While the deficiency of *mre11* did not further inhibit social interaction, this supports the idea that *mre11* is relevant to conspecifics recognition or related functions, such as odorant response, as suggested by the authors.

Reviewer #3 (Remarks to the Author):

I am happy with the edits.

Detailed Responses to Reviewers' Comments

We thank the reviewers for their thoughtful comments and suggestions which have made the manuscript stronger. We have addressed these comments as described below. The original reviews are listed point-by-point. Our responses are in blue font. Edits made in the text of the manuscript are marked in red.

Reviewer #2:

The manuscript has undergone significant improvements, and I am satisfied with the authors' response as most of my concerns have been addressed. There is only one minor point that I recommend modifying (see below). I believe there is no need for another round of review.

7. In Response to Referee's Letter (Reviewer #2), "Point 7":

*Regarding the rewritten hypothesis on how MRE11 may impact sleep as "this altered olfactory sensation enhances some social-related signals that strongly inhibit sleep in MRE11 RNAi flies," I have another concise hypothesis to explain your results: "MRE11 is essential for flies to recognize conspecifics, and the loss of MRE11 function impairs this process, such as the disruption of recognizing pheromones from conspecifics or other information through olfactory sensation. This disruption causes flies to mistake conspecifics for heterospecifics, inducing a 'nervous' or stressed state that disrupts sleep in group condition. In isolated condition, there is no such stress, so they sleep as much as wild-type flies." Importantly, this hypothesis can be easily tested by grouping *Drosophila melanogaster* with other *Drosophila* species under MRE11 knockdown or wild-type conditions, which could provide preliminary evidence. I strongly recommend, although this is an optional suggestion for this paper, that you test this hypothesis. If the results are positive, it will highlight at least two points: 1) Your multiple flies tracing system for researching the impact of heterospecifics on sleep or similar research situations. 2) The mechanism of MRE11 on sleep. This could potentially open up exciting possibilities for further investigation into the function of MRE11.*

*Reply: Thank you for this brilliant suggestion. We have now monitored each *mre11* RNAi fly and controls with another conspecific (*Drosophila melanogaster*) or a heterospecific (*Drosophila simulans*). **However, daily sleep duration is substantially***

shortened regardless of the presence of a conspecific or a heterospecific, along with decreased sleep bout duration (Supplementary Fig. 12). These results suggest that the sleep inhibition caused by mre11 deficiency is not due to mis-recognition of conspecifics as heterospecifics.

The interpretation needs to be revised, marked in purple.

I understand that the results of Fig S12 did not support the notion that sleep inhibition caused by mre11 deficiency is not due to misrecognition of conspecifics as heterospecifics. It appears that the interpretation is incorrect. In Fig S12 o-r, except for sleep latency, the sleep parameters of the three control groups of flies vary in the presence of D.m or D.s, making it challenging to claim that sleep parameters are affected by the presence of heterospecifics. This is a prerequisite for discussing whether mre11 deficiency affects sleep due to misrecognition of conspecifics as heterospecifics or not.

Considering that the sleep duration of mre11-deficient flies is even less in the presence of D.s compared to the presence of D.m (Fig S12 o,p), the authors' claim that "the sleep inhibition caused by mre11 deficiency is not due to misrecognition of conspecifics as heterospecifics" should be correct. (If sleep inhibition caused by mre11 deficiency is due to misrecognition of conspecifics as heterospecifics, the sleep duration of these flies in the presence of D.m or D.s should be comparable.)

The social interaction parameters of all control flies in Fig S12 t-u are consistent, indicating that the presence of heterospecifics suppresses the fly's social interaction control to conspecifics. While the deficiency of mre11 did not further inhibit social interaction, this supports the idea that mre11 is relevant to conspecifics recognition or related functions, such as odorant response, as suggested by the authors.

Reply: Thank you for pointing this out. We have now included additional descriptions regarding the interpretations of these results (marked in red) which read as: “However, we noticed that the frequency of social interaction is increased in mre11 RNAi flies in

the presence of a heterospecific, implying that MRE11 may be involved in conspecific/heterospecific recognition.”

Reviewer #3:

I am happy with the edits.

Reply: We are glad that our revision is satisfactory to you.